# Action boosts episodic memory encoding in humans via engagement of a noradrenergic system

Mar Yebra [1,7], Ana Galarza-Vallejo [1], Vanesa Soto-Leon [2], Javier J. Gonzalez-Rosa [1,3],
Archy O. de Berker[4], Sven Bestmann[4], Antonio Oliviero[2], Marijn C.W. Kroes[1,5] & Bryan A. Strange [1,6]

We are constantly interacting with our environment whilst we encode memories. However, how actions influence memory formation remains poorly understood. Goal-directed movement engages the locus coeruleus (LC), the main source of noradrenaline in the brain. Noradrenaline is also known to enhance episodic encoding, suggesting that action could improve memory via LC engagement. Here we demonstrate, across seven experiments, that action (Go-response) enhances episodic encoding for stimuli unrelated to the action itself, compared to action inhibition (NoGo). Functional magnetic resonance imaging, and pupil diameter as a proxy measure for LC-noradrenaline transmission, indicate increased encoding-related LC activity during action. A final experiment, replicated in two independent samples, confirmed a novel prediction derived from these data that emotionally aversive stimuli, which recruit the noradrenergic system, modulate the mnemonic advantage conferred by Go-responses relative to neutral stimuli. We therefore provide converging evidence that action boosts episodic memory encoding via a noradrenergic mechanism.

[1] Laboratory for Clinical Neuroscience, Centre for Biomedical Technology, Universidad Politecnica de Madrid, Campus Montegancedo, Madrid, Spain.
[2] Hospital Nacional de Parapléjicos, FENNSI Group, Hospital Nacional de parapléjicos Finca la Peraleda s/n 45004, Toledo, Spain. [3] University of Cadiz, Institute of Biomedical Research Cadiz (INiBICA), Puerta del Mar Hospital, Research Unit, Lab 3, 9th floor, Av. Ana de Viya, 21, 11009 Cádiz, Spain. [4] Dept Clinical and Movement Neurosciences, UCL Queen Square Institute of Neurology, University College London, 33 Queen Square WC1N3BG, London, UK.
[5] Donders Institute for Brain, Cognition, and Behaviour, Radboud University Nijmegen Medical Center, Kapittelweg 29, 6500 HB, Nijmegen, The Netherlands.
[6] Department of Neuroimaging, Alzheimer's Disease Research Centre, Reina Sofia-CIEN Foundation, Calle de Valderrebollo, 5, 28031 Madrid, Spain. [7] Present address: Cedars-Sinai 127S. San Vicente Blvd, Advanced Health Sciences Pavilion, 6th Floor, Los Angeles, CA 90048, USA. Correspondence and requests for materials should be addressed to M.Y. (email: maryegra@gmail.com)

M any of the episodic memories we form in daily life are encoded whilst we are physically active. However, the extent to which actions influence episodic memory encoding is currently unknown. Research on educational techniques shows that "active learning", an instructional method that stimulates student activity in class, such as making button-press responses, is more effective than passively receiving information from an instructor[1,2]. However, within this framework, students typically make motor responses to questions posed by the instructor, thus the effect of the action per se (button-press alone) on learning cannot be identified. A separate line of study has shown that memory for action phrases (e.g., "pick up the book") is improved when participants perform the actions during encoding compared with merely listening to or reading the phrases[3,4]. Yet, since the memory tested in this task pertains to the movement, the memory of the movement cannot be dissociated from the effect of engaging the motor system on memory. In other words, it is currently unknown whether actions influence memory for stimuli that are incidental to the movement being carried out. An example of the latter effect would be whether the likelihood of remembering the title of a book is different if we are cued to pick up the book relative to if we simply look at the same book on the library shelf.

A possible relationship between action and episodic memory is suggested by action-related neuronal responses in two brain areas: the medial temporal lobe (MTL) and the locus coeruleus (LC). MTL structures, particularly hippocampus, are critical for episodic memory and spatial navigation[5,6]. In rodents, hippocampal theta rhythmic activity has long been associated with gross voluntary types of movement such as rearing and jumping[7]. Furthermore, the activity of hippocampal place cells, which fire when the animal visits a specific area in a familiar environment[5], is also strongly dependent on movement-related information[8]. In humans, intracranial recordings from the MTL reveal that voluntary movements of the arm or tongue, in contexts not requiring explicit memory encoding, modulate neuronal firing rates in hippocampus[9] and surrounding cortex including parahippocampal gyrus[10]. These examples indicate that movement modulates neural activity in the MTL, a region critical for episodic memory, suggesting that action may affect memory formation.

The LC is the brain's main source of noradrenaline (NE), a neuromodulator known to modulate episodic memory[11–14]. Single-unit recordings of the LC in non-human primates[15–18] and cats[19] demonstrate increased activity with goal-directed actions. This raises a possibility that the NE released by action-induced LC activity may promote on-going cognitive functions, such as the encoding into episodic memory of stimuli presented simultaneously with the action. We therefore hypothesized that taking action would enhance episodic memory encoding by engaging MTL memory circuits via recruitment of the noradrenergic system.

To test this hypothesis, we examined how encoding of a visual stimulus is influenced by simultaneous voluntary movement ("Go" button-press response) compared to withholding of movement ("NoGo" response). A total of 296 healthy, young participants were tested over a series of experiments, that included functional magnetic resonance imaging (fMRI) and pupillometry studies, with different task manipulations. The experiments started with an encoding task during which participants viewed pictures with the requirement to perform an action (Go-items) or withhold an action (NoGo-items) indicated by the color of a surrounding frame. Participants subsequently performed a surprise recognition task (Fig. 1). Initial behavioral experiments confirm our prediction that action modulates memory encoding. Subsequent experiments employing fMRI, pupillometry and

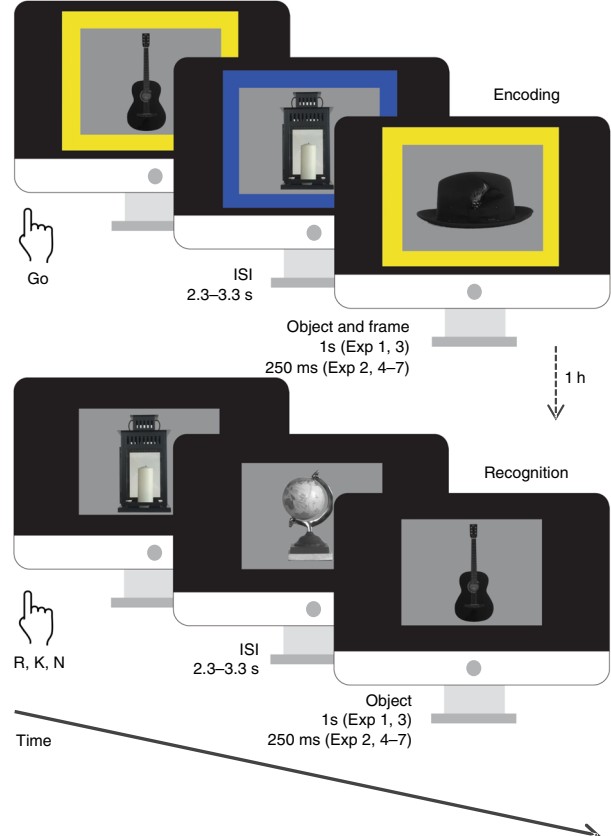

**Fig. 1** Behavioral task. Incidental memory encoding in the context of a Go/NoGo task. At encoding, grayscale objects were presented with a color fame indicating requirement of a button press for the Go condition or withholding the response for NoGo. A surprise recognition test was conducted one hour later (or one day later Exp 7), during which participants were presented with objects from the encoding task intermixed with an equal number of lure items (presented without a frame) and indicated whether they remembered (R), were familiar with (K) or did not remember (N) the objects

manipulation of the emotional content of encoded material provide converging evidence to support our hypothesis that Go-associated encoding enhancement is mediated by the adrenergic system.

## Results

**Taking action boosts episodic memory**. In the first experiment (Exp 1), we tested for the effect of performing an action during encoding on subsequent memory performance (Table 1). During the surprise recognition test, participants were required to make "remember", "know" or "new" (R/K/N) judgments[20], with remember responses indicating recall of elements of the study episode, know responses indicating a sense of familiarity, and new responses indicating the picture was not presented at encoding. We observed significantly better recollection for items requiring Go responses at encoding compared to NoGo-items (paired-sample $t$-test $t_{30} = 2.40$; $P = 0.023$), (Table 2). Successful encoding of Go items was not modulated by response speed, as reaction times (RTs) for subsequently remembered and forgotten Go items did not differ (Fig. 2b, Supplementary Table 1). By contrast, memory performance for familiarity judgments was at chance level for both Go and NoGo stimuli (one-sample $t$-tests K hits minus false alarms, $P$s > 0.49) and did not differ between them (paired-sample $t$-test $t_{30} = 0.55$; $P = 0.587$). This absence of a

**Table 1 Summary of experimental protocol**

| Exp | Presentation time | Recognition test interval | Performance financially rewarded | Context | N | Emotional stimuli |
|---|---|---|---|---|---|---|
| 1 | 1 s | 1 h | No | Behavior | 31 | No |
| 2 | 250 ms (0/250/500 ms) | 1 h | No | Behavior | 38 | No |
| 3 | 1 s | 1 h | Yes | Behavior | 26 | No |
| 4 | 250 ms | 1 h | No | Behavior | 22 | No |
| 5 | 250 ms | 1 h | No | fMRI | 21 | No |
| 6 | 250 ms | 1 h | No | Pupillometry | 28 | No |
| 7A | 250 ms | 1 day | No | Behavior | 31 | Yes |
| 7B | 250 ms | 1 day | No | Behavior | 33 | Yes |

Participants and experimental context for Exp 1–7. Presentation time pertains to both encoding and recognition tasks
N Number of participants

**Table 2 Accuracy comparison between Go and Ngo stimuli**

| Exp | t-test Go vs. NoGo | P value | Cohen's d Go vs. NoGo |
|---|---|---|---|
| 1 | $t_{30} = 2.40$ | 0.023 | 0.279 |
| 2 | $t_{37} = 3.28$ | 0.002 | 0.566 |
| 3 | $t_{25} = 2.85$ | 0.009 | 0.373 |
| 4 | $t_{21} = 2.26$ | 0.034 | 0.509 |
| 5 | $t_{20} = 1.41$ | 0.175 | 0.293 |
| 6 | $t_{27} = 2.75$ | 0.010 | 0.397 |
| 7A (neutral) | $t_{30} = 0.85$ | 0.40 | 0.172 |
| 7B (neutral) | $t_{32} = 0.43$ | 0.67 | 0.097 |

Summary of paired t-test and Cohen's d results comparing remember accuracy (% remembered items minus false alarm rate) for Go vs. NoGo stimuli for Exp 1–7

relevant memory signal for K responses, although contradicting previous literature[21], was generally the case for all subsequent experiments (Supplementary Table 2, Supplementary Fig. 2). This possibly reflects the difficulty of the memory task given that participants could be focusing more on the cue frame than on the picture. We therefore focused all further analyses on remember accuracy and its modulation by motor response at encoding. We replicate action-induced memory enhancement (AIME) of remember accuracy across six subsequent variants of this experiment (Fig. 2a, b, Table 2). The overall memory advantage conferred by making an action during encoding across experiments was assessed by a meta-analysis across these six experiments. The total random effect estimate on the difference in memory accuracy between stimuli paired with Go and NoGo trials was significant (Wilcoxon signed rank test, $z = 5.99$; $P < 0.001$) (Supplementary Fig. 1). The agreement between random and fixed effects analyses indicates the lack of heterogeneity across experiments (Test for heterogeneity, $I^2 = 0.00\%$; Cohran's $Q = 3.34$, $P = 0.65$).

**Action enhances episodic memory, inhibiting action does not impair it.** Memory was better for Go-compared to NoGo-items suggesting that taking action enhances encoding. However, as memory performance is compared between two response requirements at encoding (Go vs. NoGo), the difference in memory could alternatively be explained by response inhibition resulting in memory impairment. To test this alternative account, we utilized the fact that, due to the cue to Go or not being randomized across trials, the number of consecutive Go trials preceding NoGo trials was variable. With increasing consecutive Go trials, response inhibition mechanisms are more taxed, leading to increased commission errors (i.e., a "Go" response when "NoGo" is cued)[22,23]. By extension, an inhibitory mechanism underlying NoGo-evoked worsening of memory encoding would

predict that memory for NoGo items would decrease with increasing preceding number of Go items. We did not observe this relationship. Although participants in Exp 1 indeed showed more commission errors as a function of the number of consecutive preceding Go trials, showing a linear increase (repeated measures ANOVA with factor preceding Go trials (0, 1, 2, 3) $F_{1,30} = 5.44$; $P = 0.027$; partial $\eta^2 = 0.349$) (Fig. 2c; Supplementary Table 3), no effect of the number of preceding Go trials on NoGo-item memory was found ($F_{2.34,70.16} = 0.82$; $P = 0.487$). We note that this reasoning is derived from a null result. Since using traditional $p$ value hypothesis testing one can fail to reject the null hypothesis but the null hypothesis can never be accepted, we calculated posterior probabilities of the null hypothesis using Bayesian hypothesis testing[24]. Bayesian information criterion (BIC)-based estimation of posterior probabilities revealed $\Delta BIC_{10} = 92.04$; $Pr_{BIC}(H_0|D) \sim 1$, indicating very strong evidence in favor of the null hypothesis (Supplementary Table 3). This lack of NoGo-item memory modulation on the basis of preceding Go trials, also observed in all subsequent experiments (Fig. 2c; Supplementary Table 3), argues against an inhibitory mechanism negatively affecting memory for NoGo items.

**AIME is not dependent on stimulus presentation time window.** To provide further evidence that the mnemonic difference in memory performance between Go and NoGo-associated stimuli results from an AIME and not from a NoGo-induced (action inhibition) encoding impairment, Exp 2 tested if temporal overlap between stimulus presentation and putative inhibitory neural responses would determine memory performance. Inhibition during Go/NoGo tasks is linked to changes in the amplitude and topography of different waveforms of event-related potentials (ERPs) peaking at ~200–300 ms[25,26]. Exp 2 therefore employed a variable temporal asynchrony between cue-frame and grayscale picture presentation. In this experiment, grayscale pictures for both conditions (Go and NoGo) were presented during one of three consecutive time windows of 250 ms (0–250, 250–500, 500–750 ms; Fig. 2d) with the frame-cue presented from 0–750 ms. Importantly, an inhibitory account of NoGo-induced encoding disruption predicts that poorer NoGo than Go memory would be most pronounced in the earliest stimulus presentation window (0–250 ms), as this corresponds to the temporal profile of inhibition response-associated electrophysiological activity (e.g., the "N2 ERP" component).

Directly contradicting this account, Go item memory was better than NoGo item memory at all stimulus presentation windows (Fig. 2e), with similar RT distributions for the three windows (Fig. 2f). In a first analysis, collapsing over presentation time window, we replicate a significant effect of motor response on subsequent remember accuracy (paired-sample t-test

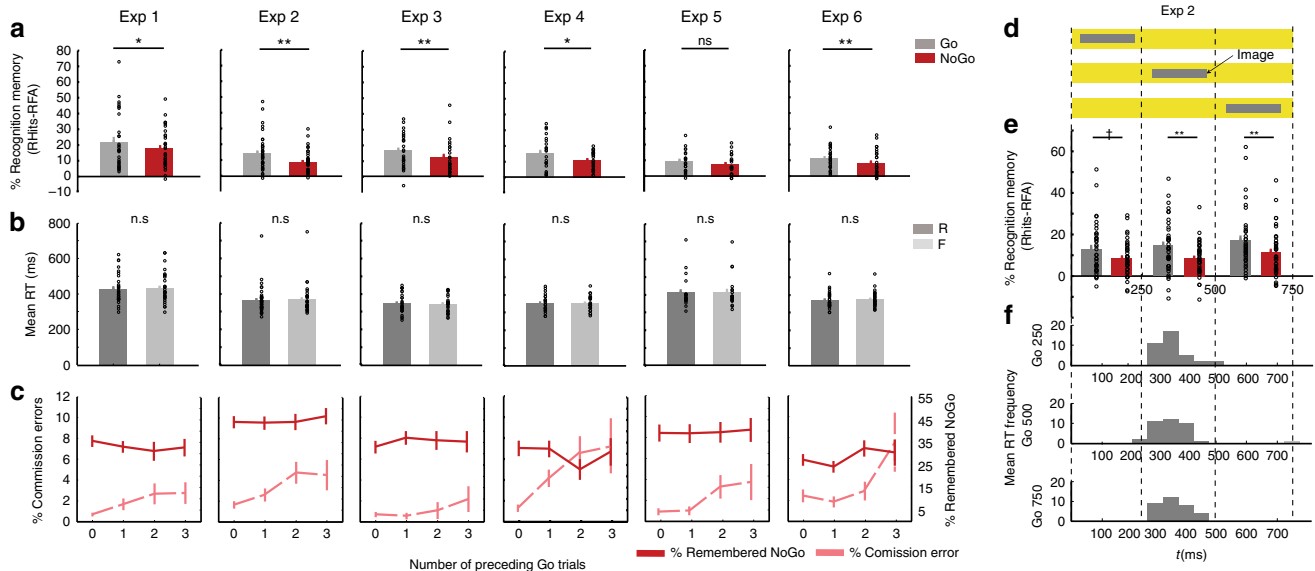

**Fig. 2** (See also Supplementary Fig. 2, Supplementary Tables 1 and 3). Behavioral results. Exp 1–6 **a** Memory performance. Recognition memory for remembered items corrected by false alarms (proportion of remembered (R) responses to new items) for both Go and NoGo conditions for each experiment *$P < 0.05$, **$P < 0.01$. **b** Reaction times at encoding for remembered and forgotten Go stimuli. **c** Commission error rate on NoGo trials during encoding depending on the number of consecutive preceding Go trials in light red and Recognition memory (remember hits minus false alarms) for the NoGo condition depending on the number of consecutive preceding Go trials in red. **d** Experimental design of Exp 2. The colored background indicating Go or NoGo response requirement was presented for 750 ms while grayscale objects were presented for 250 ms on the center of the screen in one of three possible onset times: 0, 250 or 500 ms. **e** Recognition memory for remembered items corrected by false alarms for each condition (Go/NoGo) and the time window of presentation (250/500/750 ms) †($P < 0.05$ one-tailed). **f** Histograms for the mean RTs across participants for Go condition at each time window of presentation. All error bars are the SEM. All statistical comparisons used $t$-tests

$t_{37} = 3.277$; $P = 0.002$, Fig. 2a). Critically, a repeated measures ANOVA with factors memory performance (remembered Go, NoGo) and time window of presentation (0, 250 or 500 ms) showed no significant interaction ($F_{24.615,73.334} = 0.22$; $P = 0.802$). The BIC-based estimation of posterior probabilities revealed a $\Delta BIC_{10} = 198.61$; $Pr_{BIC}(H_0|D) = \sim 1$, indicating very strong evidence in favor of the null hypothesis. This suggests that the effect of voluntary movement/withholding movement on subsequent memory does not occur exclusively at a particularly early stage of the inhibitory process (Fig. 2e). We note that Go-associated memory is greater than NoGo memory even when stimuli are presented 500 ms after cue frame onset, which is later than all RTs for this stimulus type (Fig. 2f). Although other NoGo-related ERPs occurring later than the N2 have been described, the results of Exp 2 together with the observed lack of NoGo-item memory modulation on the basis of preceding Go trials, provide evidence against inhibition-induced memory impairment.

**AIME is unlikely to reflect an effect of target detection**. Memory enhancement has been reported for task-irrelevant visual stimuli shown at the same time as a target item[27]. Target detection is typically studied in the context of low-frequency targets. We had controlled for this using a 50:50 ratio of Go:NoGo stimuli throughout all experiments. However, although the global probability of Go and NoGo is the same, the local probability varies because of the randomized presentation order. We extracted the percent subsequently remembered Go items depending on whether there were 0, 1, 2, 3 or 4 preceding NoGo items. The rationale here is that if AIME is due to a target detection process, as shown using infrequent targets[27], the increased Go-related memory should be most evident for Go items that are preceded by many NoGo stimuli (i.e., infrequent in terms of local probability). This was not the case in any of our experiments 1–6. A one-way ANOVA

on memory for Go items depending on whether there were 0,1,2,3 or 4 preceding NoGo items yielded no significant interaction for any of the experiments (Supplementary Table 4). BIC-based estimation of posterior probabilities for Exp 1 yielded a $\Delta BIC_{10} = 92.04$; $Pr_{BIC}(H_0|D) = \sim 1$ (Supplementary Table 4), showing very strong evidence in favor of the null hypothesis. We also note that target detection-evoked memory enhancement occurs when a target requires a button-press, as well as in the absence of any required action[28], suggesting that target detection modulates memory via a different mechanism than action. Furthermore, improved recognition performance for target-paired than for distractor-paired images has been shown to benefit both "familiar" and "remember" judgments[28], whereas the effect described here does not extend to familiar old judgments.

**Action enhances memory regardless of reward anticipation**. Human neuroimaging data suggest that memory formation is promoted by anticipation of reward through interactions between MTL structures and dopaminergic midbrain[29,30]. Furthermore, there is recent evidence that button-press Go responses in anticipation of reward can improve memory encoding[31]. This raises a possibility that Go responses in the current task reflect an approach-related action that engages the reward system which, in turn, strengthens episodic memory. If this were the case, the ensuing prediction is that explicit anticipation of financial reward would evoke greater action-evoked memory enhancement than observed in Exp 1. Thus, the design of Exp 3 was identical to Exp 1 (frame-cue and picture presented simultaneously) except that participants were financially rewarded for responding as fast as they could, and financially penalized for omission and commission errors. These task instructions led to significantly faster RTs for Go trials in Exp 3 than in Exp 1 (paired-sample $t$-test $t_{55} = 4.57$, $P < 0.001$). Again we demonstrate significantly better remember accuracy for Go- vs. NoGo-items ($t_{25} = 2.85$; $P =$

0.009; Fig. 2a). However, a repeated measures ANOVA on remember accuracy with within-subjects factor response type (Go vs NoGo) and between-subjects factor experiment (Exp 1 No Reward, Exp 3 Reward) did not show a significant interaction ($F_{1,55} = 0.01$; $P = 0.913$). The BIC-based estimation of posterior probabilities revealed a $\Delta BIC_{10} = 243.42$; $Pr_{BIC}(H_0|D) \sim 1$, which shows very strong evidence in favor of the null hypothesis. The main effect of Go vs. NoGo memory was significant ($F_{1,55} = 12.65$; $P = 0.001$; $\eta^2 = 0.187$), with effect sizes for the Go vs. NoGo memory comparisons comparable across the two tasks (Table 2). There was not a significant between-subject effect of Exp ($F_{1,55} = 2.65$; $P = 0.109$). These results therefore indicate that there is no additive effect of reward anticipation on the observed AIME.

**AIME is associated with increased LC activity**. Our behavioral studies were predicated on the hypothesis that taking action would boost episodic memory via interactions between the noradrenergic system and MTL memory circuits. To test this mechanistic hypothesis, we conducted a fMRI study. First, a behavioral pre-fMRI experiment (Exp 4) was performed, identical to Exp 1 but with stimulus presentation duration of 250 ms. Exp 4 simply ensured that robust memory enhancement for Go vs. NoGo stimuli is observed at this shorter presentation time, employed so as to minimize saccades, which not only lead to spurious BOLD effects[32] but also represent another type of action that increases LC activity in non-human primates[17]. The results of Exp 4 once more replicated the main finding from Exp 1–3 of better memory for Go vs. NoGo items (paired-sample t-test, $t_{21} = 2.26$; $P = 0.034$; Fig. 2b).

The task employed in the context of fMRI scanning (Exp 5) was identical to Exp 4. Behaviorally, although there was a remember advantage for Go vs. NoGo stimuli, this effect was not significant (paired-sample t-test $t_{20} = 1.41$; $P = 0.175$). The primary aim of this fMRI experiment was to derive mechanistic insights into memory enhancement for stimuli paired with action. Testing for an interaction between motor response (Go vs. NoGo) and subsequent memory (R vs. F) identified a significant activation in dorsal pons (two significant voxels), in an area consistent with LC (Fig. 3a, Supplementary Table 5). Note that this effect was also observed if the sample was restricted to the 14 subjects showing AIME. To increase the robustness of spatial localization of this response to LC, we repeated this analysis using an infra-tentorial template for spatially unbiased, nonlinear normalization of brainstem and cerebellum (SUIT) to provide more accurate intersubject-alignment of the brainstem than whole-brain methods. A significant action by subsequent memory interaction was again observed in dorsal pons. The overlap of this activation (functional image resolution of 2 mm isotropic voxels) with a probabilistic atlas of the LC (image resolution of 1 mm isotropic voxels) was nine 1 mm voxels (Fig. 3b).

Furthermore, both the Go vs. NoGo main effect comparison (Supplementary Fig. 3a, Supplementary Table 6) and the opposite test (Supplementary Fig. 3b, Supplementary Data 1), revealed cortico-subcortical networks previously shown to be involved in action and response inhibition[33,34], respectively. As predicted, MTL activation, in parahippocampal gyrus, was observed in the R vs. F, subsequent-memory comparison (Fig. 3d, f, Supplementary Table 7) as has been reported previously[35,36].

LC sends widespread noradrenergic projections to cortical and subcortical structures, including the MTL (for review see ref. [37]). To determine which regions correlate with LC activity during encoding, we performed a psychophysiological interaction (PPI) analysis to estimate context-specific changes in correlation between the LC and the rest of the brain. Specifically, we tested

which regions were functionally connected with LC under the experimental context of successful encoding between Go vs. NoGo trials. Connectivity between LC activity and parahippocampal gyrus was observed (Fig. 3e, Supplementary Table 8), in a region in close proximity with parahippocampal cortex expressing a main effect of successful object encoding (Fig. 3d). These results suggest that action-evoked memory enhancement is mediated by a LC–parahippocampal gyrus circuit: NE neuronal activity is triggered by action, as has been previously shown in animal studies[15–19], and the ensuing NE release targets the MTL promoting memory formation[11–14,38].

**AIME is associated with increased pupil dilation responses**. Our fMRI results indicate that AIME results from noradrenergic LC responses that upregulate episodic memory encoding processes in the MTL. To provide a second, independent index of LC activity during Go-induced encoding enhancement, we next performed the same Go/NoGo memory paradigm while recording pupil diameter responses (Exp 6). Pupil diameter has been shown to be a reliable, indirect index of LC activation[39], positively correlating with LC firing rates in non-human primates[40,41], and with BOLD activity in human LC[42]. Furthermore, pupil diameter is also modulated by learning and memory processes[43,44]. We therefore expected that the pupil-derived index of LC activation would relate to memory performance and action in the same way as that observed with fMRI.

The behavioral results of Exp 6 again show better remember accuracy for stimuli paired with Go vs. NoGo responses (paired-sample t-test, $t_{27} = 2.75$, $P = 0.010$) (Fig. 2a, Table 2). Note that the behavioral task used in Exp 4, 5, and 6 was identical, with Exp 4 and 6 showing significantly better remember accuracy for Go vs. NoGo stimuli. To test for an interaction between encoding and button press in pupil diameter, the raw pupil responses (Fig. 3g) were submitted to a general linear model (GLM) using a basis function approach. Stimulus-locked pupillary responses were modeled with two basis functions, one pertaining to the light-reflex and another to later cognitive component (Fig. 3h, see "Methods"). For each subject, parameter estimates for regressors convolved with the cognitive basis function pertaining to our conditions of interest (GoR, GoF, NoGoR, and NoGoF) were entered into a 2 by 2 ANOVA. Go responses evoked greater pupil dilation (main effect of Go vs. NoGo, $F_{1,27} = 17.78$; $P < 0.001$) as is clear from the raw pupil traces, where an initial pupil constriction, due to the light-reflex to stimulus presentation, is followed by a later differential dilation (Fig. 3g), in line with previous observations in human[45,46] and non-human primates[18]. Critically, a significant interaction between response type (Go vs. NoGo) and subsequent memory (R vs. F) is observed (repeated measures ANOVA $F_{1,27} = 4.21$; $P = 0.050$). As predicted from the LC activation in Exp 5, this interaction reflects greater pupil dilation to stimuli paired with Go responses that are subsequently remembered (Fig. 3i, j). These results, derived from using pupil diameter as a proxy measure of LC activation, reinforce our fMRI evidence that Go-induced encoding enhancement is mediated by the noradrenergic system.

**AIME depends on arousal**. Across all experiments we observed, at a group level of statistical inference, a consistent memory advantage for stimuli requiring a Go response at encoding. However, the advantage is not observed in all subjects. We hypothesized that this could be due to inter-subject differences in arousal levels during performance of the cognitive task. This hypothesis is based on the inverted-U relationship between arousal (and noradrenergic activity) and cognitive performance on demanding tasks (the Yerkes-Dodson law[47–49]) such as

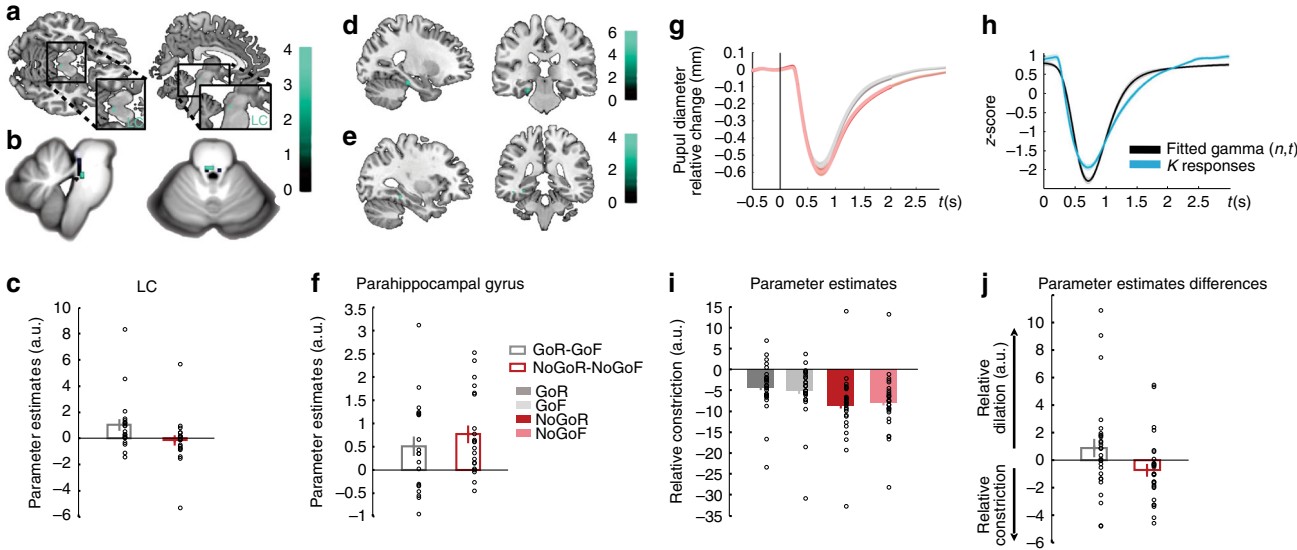

**Fig. 3** (See also Supplementary Fig. 3, Supplementary Tables 5, 6, 7, 8 and Supplementary Data 1) Go responses during successful encoding engage the noradrenergic system. **a–c** LC activation (Exp5). **a** Interaction between action (Go vs NoGo) and subsequent memory (remembered vs forgotten) has been overlaid on a canonical T1 image (threshold for illustration, here and in subsequent panels, $P < 0.001$ uncorrected) to show activation in an area of the dorsal pons consistent with LC (2 −28 −16; $Z = 3.38$; $P < 0.001$ uncorrected). **b** The same interaction, normalized to the SUIT atlas template (−2, −34, −23, $Z = 3.32$; SVC $P_{FWE} = 0.035$) is overlaid on a high resolution atlas template of the human brainstem and cerebellum, with a probabilistic spatial mask (at 1 std) of LC superimposed. **c** Parameter estimates for the BOLD response in LC. Error bars pertain to s.e.m. **d–f** Parahippocampal activation (Exp 5). **d** The comparison between remembered vs. forgotten items reveals parahippocampal activation (−24 −26 −20; $Z = 3.73$; SVC $P_{FWE} = 0.05$), shown overlaid on a canonical T1 image. **e** A psychophysiological interaction between LC activation and action-induced modulation of memory is significant in parahippocampal gyrus (−32, −38, −12; $Z = 3.79$; SVC $P_{FWE} = 0.02$). **f** Parameter estimates for the parahippocampal activation shown in (**d**). Error bars pertain to s.e.m. **g–j** Pupillary responses during successful encoding are modulated by Go responses (Exp 6). **g** Raw pupil diameter relative to baseline change measures averaged over participants (shaded error bars pertain to s.e.m.). **h** The Erlang gamma function for the early visual component (black) parameterized by fitting to the z-scored pupil responses at encoding to subsequent familiar (K) responses (blue). **i** Parameter estimates for the cognitive aspect of the pupil responses for GoR, GoF, NoGoR, and NoGoF conditions (error bars pertain to residual error of the model). **j** Parameter estimates for the differences of the relative pupil dilation/constriction for the interaction between motor response (Go, NoGo) and subsequent memory (R, F) for the cognitive aspect of the pupil dilation

episodic memory encoding in the context of a speeded Go/NoGo task. That is, if participants already show a certain degree of arousal during encoding, further Go-evoked NE release would be detrimental to encoding performance. To test this hypothesis, we performed an additional analysis on data recorded in Exp 6 to examine an index of arousal derived from pupil measures. The light-reflex (pupil constriction in response to light) shows reduced amplitude in patients with generalized anxiety[50], and is reduced in healthy individuals in the context of arousal produced by pain expectation, with this decreased light-reflex response correlating with increased subjective anxiety[51]. Our prediction was, therefore, that participants with highest level of arousal during task performance (i.e., those with reduced light reflex) would not show AIME. To confirm this prediction, we calculated the light-reflex amplitude for all participants in Exp 6 by averaging across all encoding trials for each subject. We note that measuring baseline pupil diameter would also provide an index of arousal, but we elected to measure changes in light reflex instead because the interstimulus interval in the current task may not have been sufficient for the pupil to return to baseline diameter prior to each stimulus. Figure 4 shows the pupil response as an average function for two groups: the 22 participants completing Exp 6 who showed enhanced memory for Go vs. NoGo stimuli, and the remaining six participants who did not. There is a reduction in light-reflex amplitude in the participants not showing Go-induced memory advantage compared to the participants who do show a Go-induced memory advantage (unpaired sample $t$-test $t_{10.747} = -2.23$; $P = 0.048$), supporting our explanation that these individuals have a higher level of

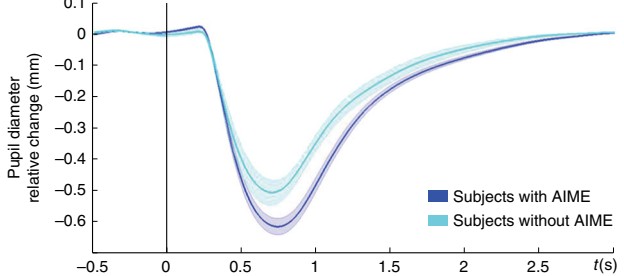

**Fig. 4** AIME and pupillary light-reflex. Reduced pupillary light-reflex constriction for participants that do not show AIME. Average pupil diameter change relative to baseline is plotted for two groups: the 22 participants completing Exp 6 who show enhanced memory for Go vs. NoGo stimuli (blue), and the remaining six participants who do not (light blue). Shaded error bars pertain to s.e.m.

arousal during encoding, which negates the mnemonic benefit afforded by Go-induced LC activity.

**AIME is modulated by emotion.** The results of Exp 5 and 6 imply involvement of LC in AIME, indicative of an underlying noradrenergic mechanism. A NE mechanism has also been shown to underlie enhanced memory for emotionally negative relative to neutral stimuli[12–14,52]. Thus, if Go- and negative emotion-induced memory enhancements are both mediated by a NE mechanism, a novel hypothesis can be derived stating that

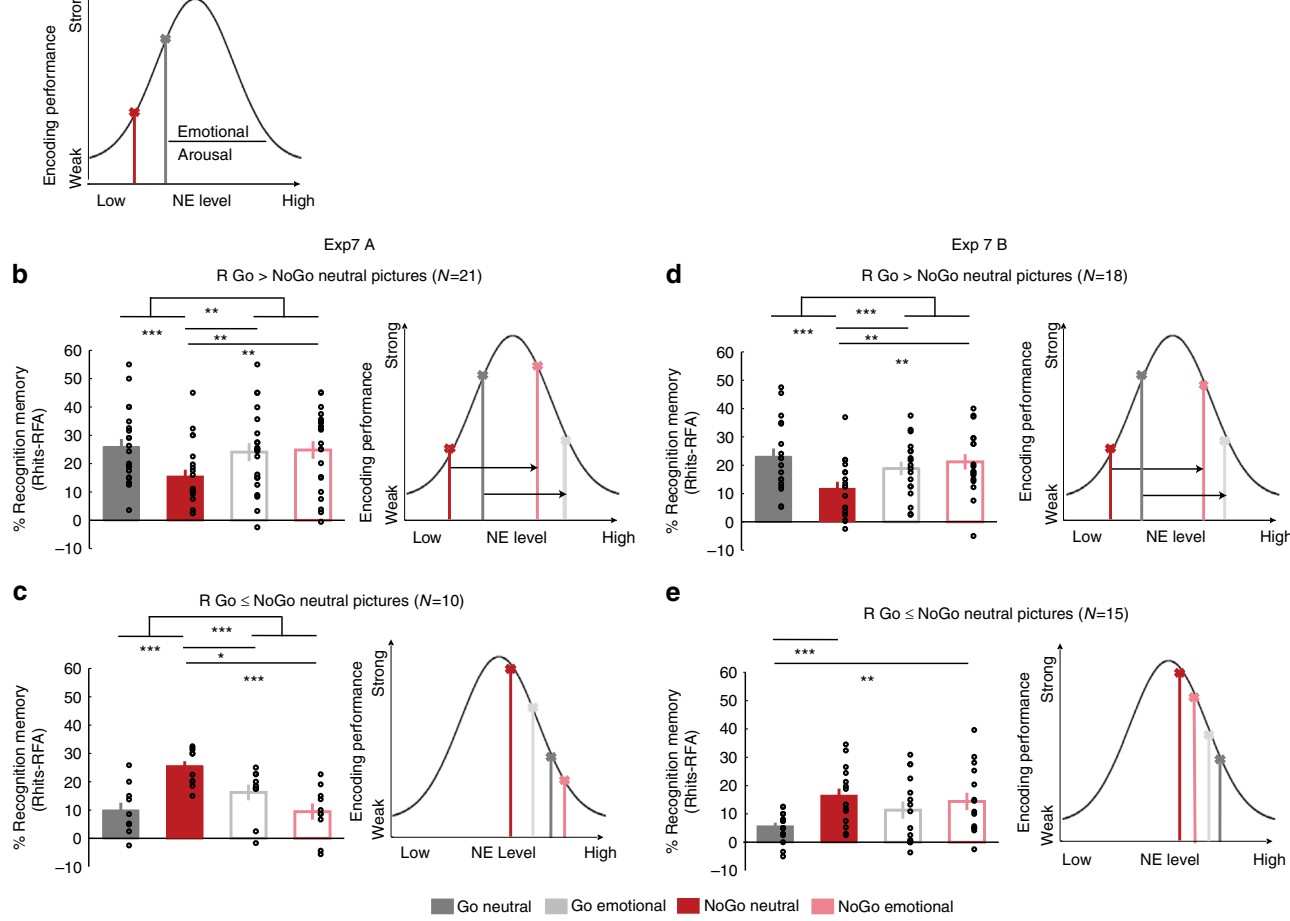

**Fig. 5** AIME is modulated by emotion. **a** Schematic "inverted-U" relationship between encoding performance and norepinephrine (NE) level, with putative locus of Go and NoGo encoding for emotionally neutral stimuli indicated on this curve. We hypothesized that emotion would shift memory scores to the right. **b**, **c** Exp 7 Recognition memory for remembered items (R) corrected for false alarm rates for Go and NoGo neutral and emotional trials (left) and the schematic (right) for participants that show AIME for the neutral stimuli (n = 21) (**b**) and those that do not (n = 10) (**c**). **d**, **e** Same as (**b**, **c**) for an independent sample replication experiment *P < 0.05, **P < 0.01, ***P < 0.001. All error bars are the SEM. All statistical comparisons used ANOVA and post hoc t-tests

AIME is modulated by the emotional nature of the stimulus presented simultaneously with movement. This rationale is again based on the inverted-U relationship between arousal (and noradrenergic activity) and cognitive performance[47]. That is, if during memory encoding there are two psychological parameters that increase noradrenergic drive (emotion and voluntary movement), these effects may influence encoding performance in a predictable way (Fig. 5a). As emotional arousal increases, encoding performance for emotional stimuli (Go emotional and NoGo emotional) should move rightwards on the inverted-U curve beyond an optimal NE effect on encoding. Highest noradrenergic drive, evoked by the condition involving both action and aversive emotion simultaneously (i.e., Go emotional) should thus worsen memory performance. Neutral stimuli (NoGo neutral, Go neutral) should lie on the left side of the curve, with lowest noradrenergic drive for the condition not involving action or aversive emotion (NoGo Neutral) showing lowest memory performance. By contrast, optimal memory performance should be situated in the middle of the curve corresponding with a moderate level of emotional arousal and NE release driven by only one NE mediator, i.e., either action or emotion (Go Neutral, NoGo Emotional) (Fig. 5a). The alternative hypothesis would be a simple summation of effects of NE drive, producing a main effect on both action and emotion, but no interaction. Indeed, it is not

the first time that different levels of LC-NE tonic functioning have been suggested to regulate other aspects of cognition including working memory. Particularly it has been recently hypothesized that there are different potential LC-NE modes explaining low working memory capacity performance: lower tonic LC activity (those that would be operating in the left side of the curve; Fig. 5a), hyperactive tonic LC activity (operating in the right side of the curve) or increased variability in LC tonic activity[53].

To test these predictions, we performed a final experiment, which we subsequently replicated (Exp 7A and B, respectively). Both were identical to Exp 4 except that, instead of grayscale pictures of objects, participants were presented with an equal number of neutral and emotional color scenes from a standardized database. The cue to Go or NoGo was again indicated by a blue/yellow frame. The enhancing effect of emotion is known to be greater when memory is tested after long (considered to be from 1 to 24 h or more) than after short immediate intervals, thus the surprise recognition test was performed after a 24 h delay to promote a greater effect of emotion on memory[54,55]. We first examined memory for participants showing AIME for neutral stimuli (21 of the 31 participants in total in Exp 7A, Fig. 5b). This subgroup was selected in view of the results of Exp 6 showing that individuals not showing AIME may already be at a heightened level of

arousal, which could obscure additional memory effects of the emotional nature of stimuli presented at encoding. Strikingly, although this subgroup of participants show Go-induced encoding enhancement for neutral stimuli, this is not observed for emotional stimuli (Fig. 5b). The Go-induced decrease in encoding of emotional pictures is in keeping with our predictions that the combination of emotion and Go-responses moved LC activity beyond the optimum of the inverted-U function for memory encoding (Fig. 5a). This effect was replicated in Exp7B on examining memory for 18 of the 33 participants showing AIME for neutral stimuli (Fig. 5d). In both Exp 7A and B, an emotion (neutral, aversive) by response (Go, NoGo) repeated measures ANOVA on encoding performance showed a signifi- cant interaction (Exp 7A: $F_{1,20} = 7.96$; $P = 0.011$, $\eta^2 = 0.285$, $P_{\text{Bootstrap1000}} = 0.0075$; Exp 7B: $F_{1,17} = 14.795$; $P = 0.001$, $\eta^2 = 0.465$, $P_{\text{Bootstrap1000}} = 0.001$) and a significant main effect of response (Exp 7A: $F_{1,20} = 5.393$; $P = 0.031$, $P_{\text{Bootstrap1000}} = 0.04$; $\eta^2 = 0.212$; Exp 7B: $F_{1,17} = 5.257$; $P = 0.035$; $\eta^2 = 0.236$, $P_{\text{Bootstrap1000}} = 0.053$), whereas the main effect of emotion was not significant. Note that a bootstrap procedure was applied to the statistical model estimation for Exp 7A and B, given that the sample of size of the replication study was based on effect size (see "Methods"). Post hoc $t$-tests revealed significantly different memory performance between Go Emotional and NoGo Neutral stimuli (Exp 7A: $t_{20} = 2.881$; $P = 0.009$; Exp 7B: $t_{17} = 3.374$; $P = 0.004$) and NoGo Neutral and NoGo Emotional stimuli (Exp 7 A: $t_{20} = -2.598$; $P = 0.017$; Exp 7B: $t_{17} = -2.790$; $P = 0.0134$). The difference between Go vs. NoGo Neutral stimulus encoding (Exp 7 A: $t_{20} = 6.622$; $P < 0.001$; Exp 7B: $t_{17} = 5.924$; $P < 0.001$) is obviously biased by preselection of participants showing this effect.

Interestingly, those participants that do not show AIME for neutral stimuli (Exp 7A: $n = 10$, Exp 7B: $n = 15$), actually show better memory for NoGo neutral pictures (Fig. 5c, e). If a Go- induced release of NE impairs memory in these participants, this would be compatible with these subjects operating more to the right of the inverted-U function of arousal. This would imply they were in a state of higher arousal than other subjects during the course of the experiment (Fig. 5c, e), in line with our findings from subjects in Exp 6 showing attenuated light-reflex. In Exp 7A, we again find an opposite pattern for Go/NoGo effects on emotional stimuli (Fig. 5c), and a significant interaction between emotion and motor response (repeated measures ANOVA $F_{1,9} = 48.171$; $P < 0.001$, $P_{\text{Bootstrap1000}} = 0.0005$; $\eta^2 = 0.843$). This inter- action was not, however, found for Exp 7B ($F_{1,14} = 2.714$; $P = 0.122$, $P_{\text{Bootstrap1000}} = 0.121$). Nevertheless, the results of Exp 7— overall—indeed confirm our predictions, based on the Yerkes- Dodson law, for memory performance showing an action–emotion interaction following an inverted-U for indivi- duals with putatively normal levels of arousal. Moreover, they provide further support for a NE basis of AIME.

## Discussion

Over a series of experiments, we consistently observed better memory for stimuli co-occurring with action. Given that in all experiments, memory for Go stimuli was compared to NoGo items, we employed two strategies to make the case that the memory difference between these two stimulus classes reflected enhanced Go, and not impaired NoGo, encoding. First, we showed that increasing inhibitory load did not disrupt successful memory encoding, despite increasing commission error rates. This contradicts a recent suggestion of response inhibition- induced episodic memory impairment[56,57]. Indeed, this previous study showed that participants committed more NoGo errors with increasing number of preceding Go stimuli, but did not test

for an expected increased disruption of memory as inhibitory load increased. We note that increasing inhibitory load with increasing consecutive Go trials increases the surprise elicited by the subsequent NoGo stimulus, and (working) memory can be impaired after surprising events that trigger motor inhibition[58]. The role of surprise, which is also associated with increased noradrenergic activity[37], is however, small in our task, given the equiprobable presentation of Go and NoGo stimuli. Second, we provide evidence that the observed memory difference did not occur exclusively at an early stage of the inhibitory process cor- responding to the temporal profile of response inhibition-evoked electrophysiological activity (e.g., the N2 ERP component)[25,26]. Both strategies indicated that an inhibitory effect of action inhi- bition is unlikely to account for the encoding difference between Go and NoGo stimuli.

Data from fMRI and pupillometry experiments provided converging evidence for LC engagement as the mechanism underlying AIME, implying a role for NE release in this process. Given the established role of NE in consolidation of memory for emotional experiences[11–14] we introduced an emotional manip- ulation to our task (Exp 7), and showed that AIME is also modulated by emotional arousal. Strikingly, the emotional con- tent of stimuli interacted with putative action-driven NE release to modulate memory performance in a way that reflects Yerkes- Dodson law. This is in keeping with early studies showing that levels of arousal interact with injected NE dose to modulate memory performance following an inverted-U curve in rats[59], meaning that low doses of NE do not alter, moderate enhance and high doses impair later retention performance. Together with our results that participants with higher levels of arousal (lower light- reflex amplitude) during task performance did not show AIME, suggests that inter-subject memory variability can be explained by arousal state. Indeed, we speculate that the absence of a significant AIME in the context of fMRI scanning (Exp 5) reflects the arousal effects of MRI scanning, known to increase sympathetic nervous system activity[60] and cortisol levels[61].

An LC-centered mechanism for AIME facilitates the inter- pretation of two further behavioral findings, when considered in the context of LC recordings in non-human primates. RTs did not differ between subsequently remembered and forgotten Go items, suggesting that the speed of response does not modulate AIME. This absence of RT-modulation mirrors findings in monkeys that the magnitude of LC firing during action is not modulated by RT[15–17]. Furthermore, in the current study, we found no additive memory benefit for financially rewarded, relative to unrewarded, Go trials. This is in line with monkey data showing LC firing aligned with action in the absence of reward anticipation; LC responses are observed with actions to no reward cues[16], on non-rewarded trials[62], with rewarded decision to act but not with rewarded decision to stop a response[17]. These observations argue in favor of the sufficiency of noradrenergic drive mediating the memory enhancement shown here. We note, however, that the evidence for engagement of LC provided here by fMRI, pupillometry and behavioral approaches, is by necessity indirect given that direct electrophysiological recordings from this area in humans is currently not possible. These findings could motivate studies in non-human animals performing direct recordings in LC during a similar experimental framework similar to the current one.

Over a series of behavioral experiments, we provide the first empirical evidence that action performance can boost episodic memory for stimuli unrelated with the movement. By dissociating the action from the content of the memory, we provide a novel dimension to the enactment effect[3,4]. In Exp 2, stimuli were presented asynchronously with the cue for motor response, lim- iting the likelihood that AIME simply reflects subjects

remembering the association between an image and the action. Furthermore, the fact that button-presses did not influence when or how visual stimuli were presented differentiates our findings from the memory benefit observed during volitional control tasks, in which participant actions lead to self-controlled viewing, as opposed to passive viewing, of encoded stimuli[63]. Furthermore, the fact that subjects did not have to choose between two alternative button press responses differentiates our findings from previous results showing that the act of choosing enhances declarative memory[64].

Our findings are supported by previous reports of enhanced memory for target-paired stimuli that require a button press[27,65,66]. This memory enhancement has been interpreted in terms of an attentional boost effect[67]. Given the critical role of the noradrenergic system[37] for attentional processes, this interpretation can be accommodated by AIME mediated by recruitment of the LC. An explanation of Go vs NoGo memory advantage on the basis of Go responses being more attentionally demanding is unlikely, given that NoGo responses also require attention, particularly in the context of financial penalization of commission errors (Exp 3). It should also be noted that studies showing enhanced memory for target-paired stimuli typically employ target-detection tasks, where the stimuli requiring action are infrequent, thereby producing an "oddball" effect which is known to improve memory[68].

Applying the mechanistic framework provided here to the everyday example given earlier, taking a book from the library shelf will trigger LC activity. The subsequent NE release will target parahippocampal gyrus to promote encoding and facilitate memory formation for the episode, including action-irrelevant aspects such as the title of the book. Thus, converging evidence presented in the current study argues for a relationship between LC and memory, proposing for the first time action as a link between them. These results provide a functional framework for potential future rehabilitation strategies using actions to enhance memory via noradrenergic engagement in individuals with memory impairment. Moreover, given the profound role of NE on cognition[37], our observations likely extend beyond the memory domain and implicate action-induced modulation of a range of cognitive processes.

## Methods

**Participants**. A total of 296 human subjects (aged 18–35; 116 female) were recruited via advertisement to participate in our study, which comprised seven experiments with one of these replicated. No individual performed more than one experiment. Participants were right- or left-handed for the behavioral experiments and all right-handed for the fMRI experiment, had no history of neurological or psychiatric disease, and normal or corrected-to-normal visual acuity. All participants provided written informed consent prior to commencement of the study. The study was approved by the ethical committee of the Universidad Politecnica de Madrid.

**Psychological task**. All experiments consisted of two phases: visual stimulus encoding in the context of a Go/NoGo task followed by a later surprise recognition test.

For Exp 1–6, from a pool of 380 grayscale photographs of objects from the Hemera Photo-Objects database, 190 were randomly selected and presented in randomized order during encoding. Participants were instructed to press a button (Go trials) when the images were presented with a particular color frame (blue or yellow) on a black background. The frame color for the "Go" and "NoGo" instruction was balanced across participants in all experiments. Go and NoGo frames appeared with equal probability (i.e., both at 50% probability). Participants were instructed to look at the center of the screen. Stimuli were displayed with 20 degrees of visual angle at a viewing distance of 60 cm. Participants returned after one hour in order to perform a surprise recognition test. For Exp 1–6, a total of 380 images—the 190 that were presented at encoding and 190 new "foils"—were presented in randomized order with no frames on a black background. Participants were required to indicate whether they remembered (R), were familiar with (K) or did not remember (new, N) the image from the encoding phase (Fig. 1).

For Exp 7, participants performed a similar Go/NoGo task but with color images selected from the International Affective Picture System (IAPS) Database[69].

We selected a total of 80 images, 40 neutral and 40 negative emotional, with the following arousal and valence ratings: emotional stimuli arousal score (SD) 6.46 (0.49) and valence score 2.27 (0.85); neutral stimuli arousal score 2.89 (0.41) and valence score 5.00 (0.24), using a scale from 1–9 with 1 being the most negative valence and 9 most arousing. At encoding, participants were presented with 20 emotional and 20 neutral images (randomly selected from the pool of 80 images). Again, images were presented in random order with a color frame indicating the requirement of pressing a button for the Go trials or not pressing for the NoGo trials. A surprise recognition phase was conducted 24 h after, instead of the 1 h interval in all previous experiments, as the enhancing effect of emotion on memory is more evident at this longer time interval[54,55]. During the recognition phase all 80 images were presented in random order. Again, participants made a Remember/Familiar/New judgment for each stimulus.

For all experiments, exclusion criteria were applied on the basis of task performance at encoding and recognition. Participants performing at less than 90% correct button press for Go, and 90% correct withheld responses for NoGo, trials were excluded from analyses. Our equal target:foil ratio allows us to define memory performance as correct hit remembered rate minus remember false alarm rate. Those participants with memory performance <0% were not further considered for analysis. In addition, participants making button-press responses for <90% of trials during recognition testing were excluded from analyses.

Note that for the Exps in which we report a significant effect on comparing recollection for Go and NoGo items (Exp 1–4 and 6), this difference remains significant (at an alpha of 0.05) even if these excluded subjects are included in the paired samples t-test. This is also the case when testing the interaction between emotion and action in Exp 7A and B (selecting subjects that showed better memory for neutral stimuli paired with Go vs. NoGo trials). This interaction remains significant including the excluded subjects (for Exp 7A, $F_{1,24} = 9.854$, $P = 0.004$, $\eta^2 = 0.291$ and for Exp 7B, $F_{1,27} = 24.444$, $P < 0.001$, $\eta^2 = 0.475$)

**Exp 1**. Image presentation time was 1 s in both phases, with variable ISI from 2.3 to 3.3 s at encoding and 2.8 to 3.3 s at recognition. A total of 33 participants (18 women; 32 right handed; age range, 18–35 years; mean age, 26.80 years; SD, 3.20) performed the experiment. Two participants were excluded on the basis of Go/NoGo task performance.

Here, and in subsequent experiments, we make reference to absence of statistically significant effects. Using traditional p value hypothesis testing one can fail to reject the null hypothesis but the null hypothesis can never be accepted. Thus, for each reported null result we additionally provide the posterior probabilities using Bayesian hypothesis testing by means of the BIC approximation approach. When comparing a null hypothesis $H_0$ with an alternative hypothesis $H_1$ the difference in BIC values can be written as:

$$\Delta BIC_{10} = n \log\left(\frac{SSE_1}{SSE_0}\right) + \log(n); \tag{1}$$

where $n$ is the number of participants, $SSE_1$ is the sums of squared errors for model $H_1$. Posterior probability for the null hypothesis $H_0$ can be written as:

$$Pr_{BIC}(H_0|D) = \frac{1}{1 + e(-\frac{1}{2}\Delta BIC_{10})} \tag{2}$$

where a posterior probability between 0.95 and 0.99 represents strong evidence and above 0.99 very strong evidence[24].

**Exp 2**. In this experiment, the image presentation time was reduced from 1 s to 250 ms. While a colored background indicating the requirement of pressing or not pressing a button was presented for 750 ms, the images of grayscale objects were presented at three different onsets relative to the 0 to 750 ms color background presentation time: at 0 s, at 250 ms and at 500 ms. The same ISI was used as in the previous experiments. Fifty-two participants performed the experiment and 38 (34 females; 48 right handed; age range, 18–35 years; mean age 28.11, SD, 4.32) participants were included in the final analysis. Five participants were rejected for poor Go/NoGo performance, two participants were rejected because of multiple button presses for the Go trials, one did not finish the task, two participants misunderstood the instructions and pressed when the image appeared not when the colored frame appeared on the screen, and four more were excluded on the basis of poor memory performance.

**Exp 3**. This was identical to Exp 1 except that participants were financially rewarded for responding as fast as they could, and financially penalized for omission and commission errors. Participants began the experiment with a 10€ voucher and were told they could earn up to 20€ on the basis of their performance. They were informed that at the end of the experiment, five Go trials and five NoGo trials would be randomly selected and for each correct NoGo, 1€ will be added to the final quantity, 0€ otherwise. For each correct Go trial they started with 1 more euro and per 0.1 s of delay above 0.5 s, 0.1€ were subtracted for the final amount. Twenty-nine participants (14 females; 28 right handed; age range, 18–35 years; mean age, 24.03; SD, 4.19) performed this experiment. One participant was discarded due to absent button-presses at recognition, and a further two participants for poor memory performance. All participants were paid 20€. The fact that all participants were paid the full 20 € raised a possibility that subjects informed one another of the fixed financial compensation. We therefore divided subjects in Exp 3 in two groups, early and late (based on order of performing the Exp) and compared RTs and commission errors between these two groups. The rationale here is that if collusion had indeed occurred, later subjects would be

slower and make more commission errors. There were no significant differences in RTs ($t_{12} = 0.96$; $P = 0.351$) or commission error rates ($t_{12} = -0.433$; $P = 0.673$).

*Exp 4.* This was identical to Exp 1 except that stimuli were presented for 250 ms. Twenty-seven participants (16 females; 26 right handed; age range, 18–35 years; mean age, 26.74; SD, 4.92) took part in this experiment. Five participants were removed from the analysis: three on the basis of Go/NoGo performance and two were excluded on the basis of low response rate during recognition.

*Exp 5.* This experiment is a replication of Exp 4 but in the context of fMRI scanning. Thirty-five participants (25 females; 35 right handed; age range, 18–35 years; mean age, 25.34; SD, 5.06) began this experiment. On the basis of stimulus-correlated head movement we excluded four participants at encoding phase, one other subject was excluded due to signal drop out in MTL, four other participants were rejected because they either made button presses to less than 85% of Go items during encoding, or withheld responses to less than 85% NoGo trials at encoding, or made less than 85% of button press responses to all stimuli during the recognition test, one other because multiple button press for the Go condition, one other for no button pressed at recognition and three were removed because of poor memory performance (Supplementary Data 2).

**fMRI data acquisition**. For each subject, a 3T Siemens Trio TIM system was used to acquire MPRAGE T1-weighted anatomical images with 1 mm$^3$ resolution (repetition time (TR), 2300 ms; echo time (TE), 2.98 ms; flip angle, 9°). During encoding, 288 gradient-echo echo-planar T2*-weighted MRI image volumes with blood oxygenation level-dependent contrast were acquired, plus five additional volumes, acquired at the start of each session and subsequently discarded, to allow for T1 equilibration effects. Each whole-brain volume comprised 40 axial slices (2.2 mm thick; distance factor 0.25; repetition time 2.43 s; echo time 30 ms; flip angle 90°, FOV 192 mm × 192 mm; matrix 64 × 64) sequentially acquired (ascending).

**fMRI data analysis**. Functional imaging data were analyzed using statistical parametric mapping (SPM8; http://www.fil.ion.ucl.ac.uk/spm) using an event-related design. Each subject's fMRI time series was realigned, slice time corrected, normalized to MNI space and smoothed with an isotropic 3D Gaussian kernel of 6 mm full-width half-maximum. To test for effects of motor action on memory, we specified six effects of interest in a general linear model (GLM): the events corresponding to Go and NoGo trials, separated according to whether these images yielded a subsequent remember (R), familiar with (K) or forgotten (F) response at recognition testing. Event-specific responses were modeled by convolving a delta-function with a canonical haemodynamic response function (HRF) to create regressors of interest. Response errors were modeled separately. Six movement parameters were modeled as nuisance covariates.

Session-specific parameter estimates of the magnitude of the hemodynamic response for each stimulus type were calculated for each voxel in the brain. A contrast of parameter estimates modeling each comparison of interest (e.g., remembered vs. forgotten Go vs. NoGo images) was calculated in a voxel-wise manner to produce, for each subject, one contrast image for that particular effect. For the random effects analysis, each subject's contrast image was entered into a one-sample t-test across participants. We report group-level analyses pertaining to the main effects and interaction term of our response (Go, NoGo) by subsequent memory (Remembered, Forgotten) 2 by 2 factorial design. In order to improve the spatial reliability of the observed LC response, the SUIT toolbox was employed, as described previously[70,71]. In brief, realigned, slice-time corrected functional images were coregistered to their subject-specific T1-weighted anatomical scan (with origin manually set at the anterior commissure). We then repeated the first level analysis described above. Again, a contrast image for the interaction term of response (Go, NoGo) by subsequent memory (Remembered, Forgotten) was calculated in a voxel-wise manner. Next, the cerebellum and brainstem were isolated in the anatomical image, and the ensuing image normalized to the SUIT atlas template using a nonlinear deformation. This deformation was then applied to the contrast image created for the interaction term, and resliced, masking out activation from outside the cerebellum or brainstem. Finally, each participant's normalized contrast images were smoothed with a 6 mm kernel and submitted to a second level GLM across subjects. We carried out a small-volume correction (SVC) to the *P* values of the ensuing maxima in LC and parahippocampal gyrus. For the latter, we used bilateral posterior parahippocampal mask from the Harvard-Oxford atlas in view of previous evidence for encoding-related responses in this area to pictures yielding subsequent high-confidence remember judgments[36]. For LC maxima, we used a probabilistic LC atlas[72] normalized to the anatomical space define by the SUIT toolbox.

*Exp 6.* This experiment was performed in the context of pupillary recordings. The behavioral task was identical to Exp 4, except for the presentation of a white fixation cross in the center of the screen for the last 500 ms of the trial to allow participants to blink. Thirty-one participants (21 females; 28 right handed; age range, 18–35 years; mean age, 27.31; SD, 5.08) performed the experiment. Two participants were excluded from analyses due to excessive number of blinks (more than 10% of trials in any one of the conditions). Another participant was excluded due to low memory performance.

**Pupil data acquisition**. The diameter of the left pupil was measured using an EyeLink 1000 System (SR Research), sampled at 1000 Hz. Participants were sat in a darkened room, and asked to maintain fixation whenever possible. A chin rest was used to minimize movement. Iso-luminance was ensured for the grayscale objects and for the blue and yellow frames separately using the Shine toolbox (www.mapageweb.umontreal.ca/gosselif/shine)[73]. Stimuli were displayed with 20 degrees of visual angle at a viewing distance of 70 cm. An analogic card of the EyeLink system was connected to an Analogic/Digital converter Cambridge Electronic Device (CED) and data acquired using Spike2 (CED) software. The data were subsequently exported in MATLAB (Mathworks, Natick, MA, USA) format and analyzed using the fieldtrip toolbox (http://www.fieldtriptoolbox.org/).

**Pupil data analysis**. Pupil diameter data were band pass filtered (0.05 to 4 Hz) using a third order Butterworth filter. Blinks were manually detected and corrected by cubic spline interpolation of samples 100 ms either side of the blink. Subsequently, visual artifact rejection was performed to remove bad interpolations. Condition-specific pupil diameter modulations were analyzed using a GLM approach as implemented previously[46]. To dissociate pupil variations due to cognitive processes vs. light-evoked pupil constrictions evoked by the appearance of visual stimuli, we modeled these two response components separately. We assumed the pupil to be a linear temporal invariant system (LTI) with an impulse response function for pupillary dilation (Diameter($t$)) described as an Erlang gamma function[74] which follows this equation as a function of time t:

$$\text{Diameter}(t) = \begin{cases} t^n e^{-nt/t_{max}}; t > 0 \\ 0; \text{otherwise} \end{cases} \qquad (3)$$

where $n$ and $t_{max}$ are parameters describing the number of layers of the system and gamma peak time, respectively. To model the cognitive pupillary response we used $n = 10.1$ and $t_{max} = 0.93$ s, the parameters estimated in Hoeks and Levelt[74]. To model the light-evoked visual response, the participant-specific pupil response to encoded stimuli evoking a subsequent familiar (K) judgment at recognition (i.e., those trials that were not included in the GLM analysis), were employed as canonical responses to estimate the parameters of interest $n$ and $t_{max}$ (using the fmincon function in MATLAB) for each participant separately (Fig. 3h).

Event-specific responses for our effects of interest (Go and NoGo remembered and forgotten trials) were modeled by convolving a delta-function with the two Erlang gamma basis functions. Nuisance regressors included: the first and second derivatives of these regressors; Go and NoGo familiar trials (convolved with the visual and cognitive response function); fixation cross presentation (convolved with visual response function); button-press RT events (convolved with the cognitive response function). The glmfit function in MATLAB was used to calculate parameter estimates for the observed (z-scored) pupil response. For each participant, the ensuing parameter estimates for the cognitive Erlang function were entered into a repeated measures ANOVA to test for a response (Go, NoGo) by subsequent memory (Remembered, Forgotten) interaction across participants.

Lastly, in order to measure the pupil constriction due to the light reflex, band-pass filtered, eye blink-corrected data were epoched into trials, baseline corrected and averaged across all trials for each participant. The maximal pupil constriction (minimum diameter) for each participant was then calculated and entered into a Welch's t-test (due to different sample sizes) comparing participants who show enhanced memory for Go relative to NoGo stimuli vs. those who do not.

*Exp 7.* Participants performing at <85% correct button press for Go, and 85% correct withheld responses for NoGo trials were excluded from analyses. Furthermore, those participants with poor memory performance for collapsed emotional and neutral stimuli or for emotional or neutral stimuli respectively (defined as correct hit remembered rate minus remember false alarm rate less than 0%) were not further considered for analysis.

Exp 7A: Thirty-eight participants (20 females; 36 right handed; age range, 18–35 years; mean age, 28.85; SD, 6.22) performed this experiment. The same presentation time and ISI as in Exp 4–6 were used for both encoding and recognition phases. Seven participants were excluded from further analysis on the basis of poor memory performance.

Exp 7B: This experiment was performed as a replication of Exp 7A. In the former subjects were included in the study until reaching an effect size of interest (interaction between emotion and action for subjects that show AIME) of at least 25%. For Exp 7B the same stop criteria for including subjects in the study was applied preserving a similar sample size as in Exp 7A. Using a stop criterion based on effect size has its limitations[24]. For this reason, we further validated the statistical robustness of our results by applying a boot-strap procedure of 1000 iterations to the memory data in Exp 7A and 7B, using the MATLAB Resampling statistical toolkit.

Fifty-one participants (32 females; 47 right handed; age range, 18–35 years; mean age, 25.56; SD, 3.85) performed this experiment. The same presentation time and ISI as in Exp 4–6 were used for both encoding and recognition phases. Three were excluded from further analysis on the basis of Go/NoGo performance and fifteen participants due to poor memory performance.

**Meta-analysis**. MedCalc statistical software (https://www.medcalc.org/) was used for the meta-analysis, calculating Fixed and random effects models, and tests for heterogeneity.

**Reporting summary**. Further information on research design is available in the Nature Research Reporting Summary linked to this article.

## Data availability

The behavioral data that support the findings of this study are available via OSF (https://osf.io/aw4e8/?view_only=c6ac94c6ce6444c689b2315572e81ac8) with the identifier DOI 10.17605/OSF.IO/AW4E8. Further imaging data generated during the current study are available from the corresponding author upon reasonable request.

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

## Acknowledgements

This work was supported by Project grants SAF2011-27766 and SAF2015-65982-R from the Spanish Ministry of Science and Innovation and Marie Curie Career Integration Fellowship (FP7-PEOPLE-2011-CIG 304248) to B.S. M.K. is supported by an H2020 Marie Sklodowska-Curie fellowship and a Society in Science-Branco Weiss fellowship. This project has received funding from the European Research Council (ERC) under the European Union's Horizon 2020 research and innovation programme (ERC-2018-COG 819814).

## Author contributions

Conceptualization, M.Y., M.K., and B.S.; Investigation, M.Y., A.G., J.G.-R., A.O. and V.S.-L.; Methodology, M.Y., B.S., A.O. de B., S.B. and M.K.; Software, M.Y.; Formal analysis, M.Y. and B.S.; Writing—Original Draft M.Y. and B.S.; Writing—Review & Editing, B.S, J. G.-R., M.K., and S.B.; Supervision and Funding Acquisition, B.S.

## Additional information

**Competing interests:** The authors declare no competing interests.

