## [Transparent Peer Review File · Nature Communications]

Reviewers' comments:

Reviewer #1 (Remarks to the Author):

In this study the authors investigate how action affect memory encoding. They performed a series of experiments including measures of behavior, brain activation and pupil size. They conclude that action boosts episodic memory encoding via engagement of noradrenergic system.

The study investigate an interesting research question using a novel approach. The paper is clearly written and very ambitious including seven experiments using different methods and measures. My major concerns are that some of the evidence are indirect, some analyses based on very small samples and that the interpretations sometimes are a little speculative.

I think that the behavioral experiments 1-4 are sound and show a robust effect of action on memory performance. The only issue is that there are many subject's data in experiment 2 that could not be used (i.e. >25% of subject data were discarded).

Also in the fMRI experiment (exp 5) more than 40% of the subject data are discarded. This is very problematic and should be addressed and preferably corrected for in an independent experiment. Apart from this is the methods are sound. The results show activations in the region of Locus Coeruleus. However, the go vs nogo conditions show no difference in memory performance and the behavioral results are not as robust as suggested in exp 1-4. Thus, the results are indirect (i.e. not saying anything direct on noradrenaline), the behavioral findings not significant and not perfectly reliable due to the large discarding of data.

In exp 6 the authors are using measure of the pupil size as an index of firing of noradrenaline neuron firing. Again, the authors rely on an indirect measure but the method seem to be sound and valid.

In exp 7 the authors investigate the role of arousal. The authors rely on subject's rating of arousal. Why not using measures such as electrodermal activity? Also, trends are reported and very small sample sizes are used in some of the analyses and the interpretation end up to be a little speculative.

In summary, I find the study to be very ambitious and interesting but the measures are indirect and some of the analyses/interpretations are a little weak. I would suggest to make another experiment using fMRI and measuring pupil size simultaneously to make interpretations more reliable. I am not sure that arousal needs to be addressed in the paper but could be used in a follow-up study where I would suggest to also include a psychophysiological measure.

Reviewer #2 (Remarks to the Author):

This paper reports on seven experiments testing the hypothesis that, because goal-directed movement engages the LC, participants should have better incidental memory for stimuli presented during trials in which they made a button press (go trials) than on trials without a button press (no-go trials). Five of the seven experiments yielded significantly better memory later for stimuli from go than from no-go trials. In addition, there was a small cluster overlapping with the LC that showed an interaction (when tested at small volume correction levels) between response and subsequent memory, and pupil dilation response showed the same interaction. These are quite neat findings and the multiple replications of the key effect bolsters the case that it is a reliable effect. I believe this could meet the high threshold for a Nature Communications paper if the authors are able to address the concerns I had, detailed below.

Previous work by Aaron Seitz and others has demonstrated that memory is enhanced for task-irrelevant visual stimuli shown at the same time as a target item. The current paper goes beyond these previous findings by making the case that taking action boosts episodic memory because taking action stimulates the LC, which in turn enhances episodic memory via release of NE modulating encoding processes.

As the authors discuss, unlike previous studies showing that irrelevant information is encoded better during target detection, the current experiments involved equal proportions of targets and non-targets. However, this does not eliminate the possibility that the effects are due to target detection rather than to taking action. Given the claim that the effects stem from taking action, it seems important to disentangle target detection and action in the experimental design. For instance, the participants could be instructed with auditory or written commands on each trial what to do: e.g., 1) press button under R finger; 2) do nothing; 3) press button under L finger. This would involve taking action or not taking action but would not involve target detection.

The authors make a complex case for the results in Experiment 7, in which they posit that there should be an inverted-U relationship. They cite Yerkes and Dodson (1908) for the hypothesis that there should be an inverted-U relationship in terms of the amount of arousal and its effect on memory encoding. To make the case for an inverted-U, more needs to be done than just cite Yerkes and Dodson (1908), one of the most mis-cited papers in psychology (see Diamond, Campbell, Park, Halonen, & Zoladz, 2007 for discussion). The only other paper cited for this inverted-U case was Gold et al., (1997), who used epinephrine injections in mice posttraining, and found non-straightforward interactions of delay between training and injection and amount of epinephrine injected. A more direct basis in the literature is needed to make the inverted-U case here. Currently, without replication in a separate sample, the division of the participants into 2 groups seems post-hoc.

Other comments

How many voxels were significant in the LC cluster presented in Figure 3 and how many overlapped with the LC atlas template ROI?

For the pupil dilation experiment, it is hard to map the reported significant interaction of response type and subsequent memory onto Figure 3i, as the mean in each condition is not clear from the figure.

Were pupil measures collected in any of the other experiments? It would have been helpful to collect them in all experiments.

The neutral condition from Experiment 7 should be reported on in Table 2, with all 21 participants included. Currently it is not clear if the effect replicated in this experiment without omitting those participants who did not show the effect.

Were any participants in more than one of these experiments?

There were a high number of excluded participants, especially for the fMRI study.

On p. 31, it is reported that "Up to 52 participants performed the experiment". Why not report the exact number?

On p. 32, it is stated that all participants were paid 20 Euros. Does this mean that the performance-based payments the participants were instructed about were not implemented? If so, this deception might have been reported to friends participating in the study, reducing incentives to aim for those stated rewards. Or was it that everyone made it to the maximal payment value?

On p. 32, I did not follow the statement, "6 other participants were rejected because they presented less than 85% of correct Go or NoGo trials or answers at recognition phase"

It would be helpful if the authors would address whether their data and materials are ready to be made available in a publicly accessible online repository such as Open Science Framework upon publication, with data from excluded participants included in the files. Making the data available allows other scientists to reproduce the analyses and making the materials available allows for replications with new samples. This is especially important for potentially high profile papers like the present one.

Have all measures and tasks the participants completed been reported?

Response to reviewers' comments

“Action boosts episodic memory encoding in humans via engagement of a noradrenergic system” NCOMMS-17-25548

We thank both reviewers for their thorough and constructive evaluation of our manuscript. Below, we provide a point-by-point rebuttal (in italics) to all comments raised. Additions to the manuscript are shown in red.

Reviewer #1 (Remarks to the Author):

In this study the authors investigate how action affect memory encoding. They performed a series of experiments including measures of behavior, brain activation and pupil size. They conclude that action boosts episodic memory encoding via engagement of noradrenergic system.

The study investigate an interesting research question using a novel approach. The paper is clearly written and very ambitious including seven experiments using different methods and measures.

We thank the reviewer for his/her positive appraisal of our work.

1. My major concerns are that some of the evidence are
 - a. indirect

We agree that the evidence for engagement of human LC is – by necessity – indirect, given that direct electrophysiological recordings from this brain area in humans is (currently) not possible. For this reason, we used a combination of techniques (fMRI, pupillometry and behavioral manipulation introducing emotional stimuli) to provide converging evidence to support our inference for involvement of LC in action-induced memory enhancement. However, we now make explicit reference to the indirect nature of our observations in the discussion, and suggest that our findings could motivate studies in non-human primates

performing single unit recordings during a similar task combining Go-NoGo instructions and episodic memory encoding. (page 19).

“We note, however, that the evidence for engagement of LC provided here by fMRI, pupillometry and behavioral approaches, is by necessity indirect given that direct electrophysiological recordings from this area in humans is currently not possible. These findings could motivate studies in non-human animals performing direct recordings in LC during a similar experimental framework as here.”

b. some analyses based on very small samples

The reviewer makes a fair point regarding sample size, which we interpret pertains to Exp 7. To address this concern (also raised by Reviewer 2 point 3), we have now increased our sample size until an effect size (η^2) of at least 25% was reached for Exp 7 (now referred to as Exp 7A) and performed an independent replication of Exp 7 (Exp 7B) applying the same criteria and preserving a similar sample size as in Exp 7 A (see Online Methods, page 9-10).

“Exp 7 B: This experiment was performed as a replication of Exp 7 A. In the former subjects were included in the study until reaching an effect size of interest (interaction between emotion and action for subjects that show AIME) of at least 25%. For Exp 7 B the same stop criteria for including subjects in the study was applied preserving a similar sample size as in Exp 7 A.” *(see Online Methods, page 9-10).*

Effects that were at trend level significance in Exp 7 in our original submission are now significant with the inclusion of more subjects. (Pages 14-17. Figure 5).

“... we performed a final experiment, which we subsequently replicated (Exp 7A and B, respectively). Both were identical to Exp 4 except that, instead of grayscale pictures of objects, participants were presented with an equal number of neutral and emotional color

scenes from a standardized database. The cue to Go or NoGo was again indicated by a blue/yellow frame. The enhancing effect of emotion is known to be greater when memory is tested after long (considered to be from 1h to 24h or more) than after short immediate intervals, thus the surprise recognition test was performed after a 24h delay to promote a greater effect of emotion on memory^{51,52}. We first examined memory for participants showing Go-induced memory enhancement for neutral stimuli (21 of the 31 participants in total in Exp 7 A, Figure 5b). This subgroup was selected in view of the results of Exp 6 showing that individuals not showing action-induced memory enhancement may already be at a heightened level of arousal, which could obscure additional memory effects of the emotional nature of stimuli presented at encoding. Strikingly, although this subgroup of participants show Go-induced encoding enhancement for neutral stimuli, this is not observed for emotional stimuli (Figure 5b). The Go-induced decrease in encoding of emotional pictures is in keeping with our predictions that the combination of emotion and Go-responses moved LC activity beyond the optimum of the inverted-U function for memory encoding (Figure 5a). This effect was replicated in Exp7 B on examining memory for 18 of the 33 participants showing Go-induced memory enhancement for neutral stimuli (Figure 5d). In both Exp 7A and B, an emotion (neutral, aversive) by response (Go, NoGo) ANOVA on encoding performance showed a significant interaction (Exp 7 A: $F_{1,20} = 7.96$; $P = 0.011$, $\eta^2 = 0.285$; Exp 7 B: $F_{1,17} = 14.795$; $P = 0.001$, $\eta^2 = 0.465$) and a significant main effect of response (Exp 7 A: $F_{1,20} = 5.393$; $P = 0.031$; $\eta^2 = 0.212$; Exp 7 B: $F_{1,17} = 5.257$; $P = 0.035$; $\eta^2 = 0.236$), whereas the main effect of emotion was not significant. Post-hoc t -tests revealed significantly different memory performance between Go Emotional and NoGo Neutral stimuli (Exp 7 A: $t_{20} = 2.881$; $P = 0.009$; Exp 7 B: $t_{17} = 3.374$; $P = 0.004$) and NoGo Neutral and NoGo Emotional stimuli (Exp 7 A: $t_{20} = -2.598$; $P = 0.017$; Exp 7 B: $t_{17} = -2.790$; $P = 0.0134$). The difference between Go vs. NoGo Neutral stimulus encoding (Exp 7 A: $t_{20} = 6.622$; $P < 0.001$; Exp 7 B: $t_{17} = 5.924$; $P < 0.001$) is obviously biased by preselection of participants showing this effect.

Interestingly, those participants that do not show Go-induced memory enhancement for neutral stimuli (Exp 7 A: $N = 10$, Exp 7 B: $N = 15$), actually show better memory for NoGo

neutral pictures (Figure 5c, e). If a Go-induced release of NE impairs memory in these participants, this would be compatible with these subjects operating more to the right of the inverted-U function of arousal, *i.e.*, they were in a state of higher arousal than other subjects during the course of the experiment (Figure 5c, e), in line with our findings from subjects in Exp 6 showing attenuated light-reflex. In Exp 7A, we again find an opposite pattern for Go/NoGo effects on emotional stimuli (Figure 5c), and a significant interaction between emotion and motor response ($F_{1,9} = 48.171$; $P < 0.001$; $\eta^2 = 0.843$). This interaction was not, however, found for Exp 7 B ($F_{1,14} = 2.714$; $P = 0.122$). Nevertheless, the results of Exp 7 – overall – indeed confirm our predictions, based on the Yerkes-Dodson law, for memory performance showing an action-emotion interaction following an inverted-U for individuals with putatively normal levels of arousal. Moreover, they provide further support for a NE basis of action-induced memory enhancement.”

In the Methods section, we now provide the updated subject numbers for Exp 7A and B (Page 9-10)

“ **Exp 7.**

Participants performing at less than 85% correct button press for Go, and 85% correct withheld responses for NoGo, trials were excluded from analyses. Furthermore, those participants with poor memory performance for collapsed emotional and neutral stimuli or for emotional or neutral stimuli respectively (defined as correct hit remembered rate minus remember false alarm rate less than 0%) were not further considered for analysis.

Exp 7 A: 38 participants (20 females; 36 right handed; age range, 18–35 years; mean age, 28.85; SD, 6.22) performed this experiment. The same presentation time and ISI as in Exp 4-6 were used for both encoding and recognition phases.

One was excluded from further analysis on the basis of Go/NoGo performance and six participants due to poor memory performance.

Exp 7 B: This experiment was performed as a replication of Exp 7 A. In the former subjects were included in the study until reaching an effect size of interest (interaction between emotion and action for subjects that show AIME) of at least 25%. For Exp 7 B the same stop criteria for including subjects in the study was applied preserving a similar sample size as in Exp 7 A. 51 participants (32 females; 47 right handed; age range, 18–35 years; mean age, 25.56; SD, 3.85) performed this experiment. The same presentation time and ISI as in Exp 4-6 were used for both encoding and recognition phases.

Nine were excluded from further analysis on the basis of Go/NoGo performance and fourteen participants due to poor memory performance.”

Figure 5. Go-induced encoding enhancement is modulated by emotion. (a) Schematic “inverted-U” relationship between encoding performance and norepinephrine (NE) level, with putative locus of Go and NoGo encoding for emotionally neutral stimuli indicated on this curve. We hypothesized that emotion would shift memory scores to the right. (b-c) Recognition memory for remembered items (R) corrected for false alarm rates for Go and NoGo neutral and emotional trials (left) and the schematic (right) for participants that show Go-induced memory enhancement for the neutral stimuli ($N = 16$) (b) and those that do not ($N = 5$) (c) * ($P < 0.05$), ** ($P < 0.01$), * ($P < 0.001$).**

c. and that the interpretations sometimes are a little speculative.

Throughout the discussion we now explicitly state where our interpretation extends beyond the current data and provide examples of future studies to confirm speculative interpretation. (Pages 18, 19)

“... Indeed, we speculate that the absence of a significant action-induced memory enhancement in the context of fMRI scanning (Exp 5) reflects the arousal effects of MRI scanning, known to increase sympathetic nervous system activity⁵⁵ and cortisol levels⁵⁶. ...”

“... We note, however, that the evidence for engagement of LC provided here by fMRI, pupillometry and behavioral approaches, is by necessity indirect given that direct electrophysiological recordings from this area in humans is currently not possible. These findings could motivate studies in non-human animals performing direct recordings in LC during a similar experimental framework as here.”

2. I think that the behavioral experiments 1-4 are sound and show a robust effect of action on memory performance. The only issue is that there are many subject's data in experiment 2 that could not be used (i.e. >25% of subject data were discarded).

The incidental nature of the encoding task (i.e., the recognition test was a surprise) and brief presentation time make this a difficult memory task so several subjects performed below chance at recognition. The presentation time was reduced from 1 s to 250 ms to enable staggering of object picture presentation with respect to Go/NoGo cue. This shorter presentation time – which also reduced memory performance relative to 1s presentation – was maintained for the functional MRI study to eschew saccade-related confounds.

As stated in response to Reviewer 2, point 9, data from all excluded subjects (and the reason for their exclusion) is provided in the database of our behavioral results to be made publicly available, and added to our revision for consideration for the reviewers. Note that for the Exps in which we report a significant effect on comparing recollection for Go and NoGo items (Exp 1-4 and 6), this difference remains significant (at an alpha of 0.05) even if these excluded subjects are included in the paired samples t-test. This is also the case when testing

the interaction between emotion and action in Exp 7 A and B (selecting subjects that showed better memory for neutral stimuli paired with Go vs. NoGo trials). This interaction remains significant including the excluded subjects (for Exp 7A, $F_{1,24} = 9.854$, $P = 0.004$, $\eta^2 = 0.291$ and for Exp 7B, $F_{1,27} = 24.444$, $P < 0.001$, $\eta^2 = 0.475$).

This might suggest that our effect sizes are sufficient to reveal the effect even if these subjects are included. However, in the tasks used here, it only makes sense to ask if action has an effect on memory, if a sufficient memory performance can be shown in the first place. For this reason, we would argue that those subjects should be excluded – transparently – as done here.

We now make reference to these effects in the methods section (P 2-3)

3. Also in the fMRI experiment (exp 5) more than 40% of the subject data are discarded. This is very problematic and should be addressed and preferably corrected for in an independent experiment. Apart from this is the methods are sound. The results show activations in the region of Locus Coeruleus. However, the go vs nogo conditions show no difference in memory performance and the behavioral results are not as robust as suggested in exp 1-4. Thus, the results are indirect (i.e. not saying anything direct on noradrenaline), the behavioral findings not significant and not perfectly reliable due to the large discarding of data.

As mentioned above in response to point 2, significant Go vs. NoGo memory effects reported in our original submission remain significant when excluded subjects are introduced into the statistical comparison. Go-induced memory enhancement in Exp 5, conducted in the context of MRI scanning, remains the exception. Our explanation for this is that MRI scanning is known to increase sympathetic nervous system activity¹ and cortisol levels². We show in the current manuscript that subjects that are generally more aroused (i.e., those showing reduced light reflex – Exp 6) show less action-induced memory enhancement. Thus, even if we were to repeat the fMRI experiment, it remains possible that the AIME would not be as robust as under less stressful conditions.

The following has been added to the discussion section:

“... Indeed, we speculate that the absence of a significant action-induced memory enhancement in the context of fMRI scanning (Exp 5) reflects the arousal effects of MRI scanning, known to increase sympathetic nervous system activity⁵⁵ and cortisol levels⁵⁶. ...”

(Page 18)

4. In exp 6 the authors are using measure of the pupil size as an index of firing of noradrenaline neuron firing. Again, the authors rely on an indirect measure but the method seem to be sound and valid.

We are pleased that the reviewer finds our method here sound and valid. As outlined in responses to point 1a, given that direct electrophysiological recordings cannot currently be made from human LC, we adopted several indirect measures to support our claim of LC involvement in AIME.

5. In exp 7 the authors investigate the role of arousal. The authors rely on subject's rating of arousal. Why not using measures such as electrodermal activity?

The reviewer makes a nice suggestion. We had considered measuring skin conductance, but elected for pupil dilation in view of what we interpret as stronger evidence for the relationship between LC activity and pupil dilation in humans³ and non-human primates⁴. We also point out that pupillary changes during picture viewing have been shown to covary with skin conductance change^{5,6}.

Also, trends are reported and very small sample sizes are used in some of the analyses and the interpretation end up to be a little speculative.

We have now increased our sample size until an effect size (η^2) of at least 25% was reached for Exp 7 (now referred to as Exp 7A) and performed an independent replication of Exp 7 (Exp 7B) applying the same criteria and preserving a similar sample size as in Exp 7 A (see Online Methods, page 9). Thus, providing more robust results for memory performance

showing an action-emotion interaction following an inverted-U for individuals with putatively normal levels of arousal (Pages 14, 15, 16, 17. Figure 5).

6. In summary, I find the study to be very ambitious and interesting but the measures are indirect and some of the analyses/interpretations are a little weak.

To reassure the reviewer of the robustness of the action-induced memory enhancement we report, we have performed a meta-analysis over our individual experiments. The following has been added to the manuscript (Page 6)

“The overall memory advantage conferred by making an action during encoding across experiments was assessed by a meta-analysis across these 6 Exp. The total random effect estimate on the difference in memory accuracy between stimuli paired with Go and NoGo trials was significant ($z = 5.99$; $P < 0.001$) (Supplementary Figure 1). The agreement between random and fixed effects analyses indicates the lack of heterogeneity across experiments ($I^2 = 0.00\%$; Cohran's $Q = 3.34$, $P = 0.65$).”

We have also included a new Figure in the supplementary material to illustrate this meta-analysis result.

Supplementary Figure 1. Results of meta-analysis across experiments. The differences in mean memory accuracy between stimuli paired with Go vs. NoGo responses for Exp 1 to 6 is plotted with 95% confidence intervals (total random effect estimate of 3.57; $z = 5.99$; $P < 0.001$). Pooled effects (random and fixed) are represented by a diamond, the location of the diamond representing the estimated effect size and the width representing the precision of the estimate. Note that random and fixed models agree when there is no heterogeneity, *i.e.*, no variation in outcomes between experiments ($I^2 = 0.00\%$; Cohran's $Q = 3.34$, $P = 0.65$).

I would suggest to make another experiment using fMRI and measuring pupil size simultaneously to make interpretations more reliable. I am not sure that arousal needs to be addressed in the paper but could be used in a follow-up study where I would suggest to also include a psychophysiological measure.

The reviewer makes some excellent suggestions for follow-up studies to further characterize action-induced memory enhancement, for which we now provide evidence from 8 separate experiments. As mentioned above, pupil responses have been shown to covary with skin conductance change during emotional picture viewing so it would be interesting to show that this extends to action-induced pupil changes and memory encoding enhancement.

Simultaneous fMRI and pupillometry is currently not feasible with our scanning facility (we would need to purchase an MRI-compatible eye-tracker).

Reviewer #2 (Remarks to the Author):

This paper reports on seven experiments testing the hypothesis that, because goal-directed movement engages the LC, participants should have better incidental memory for stimuli presented during trials in which they made a button press (go trials) than on trials without a button press (no-go trials). Five of the seven experiments yielded significantly better memory later for stimuli from go than from no-go trials. In addition, there was a small cluster overlapping with the LC that showed an interaction (when tested at small volume correction levels) between response and subsequent memory, and pupil dilation response showed the same interaction. These are quite neat findings and the multiple replications of the key effect bolsters the case that it is a reliable effect. I believe this could meet the high threshold for a Nature Communications paper if the authors are able to address the concerns I had, detailed below.

We are very grateful to Reviewer #2 for her insightful and constructive criticisms and generally positive appraisal of our submission.

1. Previous work by Aaron Seitz and others has demonstrated that memory is enhanced for task-irrelevant visual stimuli shown at the same time as a target item. The current paper goes beyond these previous findings by making the case that taking action boosts episodic memory because taking action stimulates the LC, which in turn enhances episodic memory via release of NE modulating encoding processes. As the authors discuss, unlike previous studies showing that irrelevant information is encoded better during target detection, the current experiments involved equal proportions of targets and non-targets. However, this does not eliminate the possibility that the effects are due to target detection rather than to taking action. Given the claim that the effects stem from taking action, it seems important to disentangle

target detection and action in the experimental design. For instance, the participants could be instructed with auditory or written commands on each trial what to do: e.g., 1) press button under R finger; 2) do nothing; 3) press button under L finger. This would involve taking action or not taking action but would not involve target detection.

The reviewer makes the interesting suggestion that the action-induced memory enhancement we report may be due to a target detection effect. We expressed our concerns regarding the experimental approach suggested by the reviewer (see copies of email exchanges between the senior author and Reviewer #2 below). Specifically, we were concerned that even if we added another stimulus type requiring a Go response (by having stimuli cueing left-hand button presses in addition to right-hand presses), these stimuli could still be considered "targets" relative to stimuli associated with a cue to NoGo. Moreover, looking deeper into the target detection and memory literature, there are some important findings that indicate that this effect differs from what we report. Target detection-evoked memory enhancement occurs when a target requires a button-press, as well as in the absence of any required action⁷. Furthermore, improved recognition performance for target-paired than for distractor-paired images has been shown to benefit both "familiar" and "remember" judgments⁸, whereas the effect described in our work did not extend to familiar old judgments.

Nevertheless, we have now tried different experimental approaches to address the reviewer's comment. Our first approach is based on the fact that target detection is typically studied in the context of low-frequency targets. We had controlled for this using a 50:50 ratio of Go:NoGo stimuli throughout all experiments. However, although the global probability of Go and NoGo is the same, the local probability varies because of the randomized presentation order. We now provide evidence that local probability does not modulate action-induced memory enhancement. The results of these analyses have been added to the results section (Pages 8 and 9).

Approach 1.

We calculated the encoding success of Go stimuli depending on the number of preceding NoGo stimuli. Even if a Go stimulus occurs after multiple consecutive NoGo items (rendering the stimulus an infrequent target), there is no difference in memory compared to Gos after fewer or no preceding NoGos

The following has been added to the results section (Page 8-9), which makes reference to Supplementary Table 4

“Action-induced memory enhancement is unlikely to reflect an effect of target detection.

Memory enhancement has been reported for task-irrelevant visual stimuli shown at the same time as a target item. Target detection is typically studied in the context of low-frequency targets. We had controlled for this using a 50:50 ratio of Go:NoGo stimuli throughout all experiments. However, although the global probability of Go and NoGo is the same, the local probability varies because of the randomized presentation order. We now provide evidence that local probability does not modulate action-induced memory enhancement. We extracted the percent subsequently remembered Go items depending on whether there were 0, 1, 2, 3 or 4 preceding NoGo items. The rationale here is that if Go-induced memory enhancement is due to a target detection process, as shown using infrequent targets²⁶, the increased Go-related memory should be most evident for Go items that are preceded by many NoGo stimuli (*i.e.*, infrequent in terms of local probability). This was not the case in any of our experiments 1-6. A one-way ANOVA on memory for Go items depending on whether there were 0,1,2,3 or 4 preceding NoGo items yielded no significant interaction for any of the experiments (Supplementary table 4). We also note that target detection-evoked memory enhancement occurs when a target requires a button-press, as well as in the absence of any required action²⁷, suggesting that target detection modulates memory via a different mechanism than action. Furthermore, improved recognition performance for target-paired than for distractor-paired images has been shown to benefit both “familiar” and “remember” judgments²⁷, whereas the effect described here does not extend to familiar old judgments.”

Supplementary Table 4. One way ANOVA on memory for Go items depending on whether there were 0,1,2,3 or 4 preceding NoGo item for experiments from 1 to 6.

Exp	One Way ANOVA on memory performance
1	$F_{1.98, 59.53} = 1.72 ; P = 0.188$
2	$F_{1.40, 52} = 0.61 ; P = 0.49$
3	$F_{2.2, 55} = 0.36 ; P = 0.72$
4	$F_{2.05, 42.98} = 1.63 ; P = 0.21$
5	$F_{2.05, 41.07} = 0.16 ; P = 0.85$
6	$F_{1.85, 49.94} = 0.18 ; P = 0.82$

In the second and third approaches, we have performed a further 3 experiments (Exp 8a, b and c), testing a total of 104 subjects (93 included under our exclusion criteria), based on the original suggestion of the reviewer (Exp 8c) and on ideas generated from discussion with the reviewer (Exp 8a and b). All three Exps involved the inclusion of two types of action, as well as response inhibition.

[Redacted]

In general, we now believe that the act of having to select between two actions (or inaction) is interacting with our main finding in interesting, but unpredictable, ways. We would therefore prefer to reserve the results of Exp 8A-C for a future publication on these action selection effect.

2. The authors make a complex case for the results in Experiment 7, in which they posit that there should be an inverted-U relationship. They cite Yerkes and Dodson (1908) for the hypothesis that there should be an inverted-U relationship in terms of the amount of arousal and its effect on memory encoding. To make the case for an inverted-U, more needs to be done than just cite Yerkes and Dodson (1908), one of the most mis-cited papers in psychology (see Diamond, Campbell, Park, Halonen, & Zoladz, 2007 for discussion). The only other paper cited for this inverted-U case was Gold et al., (1997), who used epinephrine injections in mice posttraining, and found non-straightforward interactions of delay between training and injection and amount of epinephrine injected. A more direct basis in the literature is needed to make the inverted-U case here.

Our apologies for the superficial referencing to the literature regarding this reported effect¹¹⁻¹³.

We now state on Page 13: “This hypothesis is based on the inverted-U relationship between arousal (and noradrenergic activity) and cognitive performance on demanding tasks (the Yerkes-Dodson law⁴⁵⁻⁴⁷) such as episodic memory encoding in the context of a speeded Go NoGo task.”

3. Currently, without replication in a separate sample, the division of the participants into 2 groups seems post-hoc.

We have now increased our sample size until an effect size (η^2) of at least 25% was reached for Exp 7 (now referred to as Exp 7A) and performed an independent replication of Exp 7 (Exp 7B) applying the same criteria and preserving a similar sample size as in Exp 7 A (see Online Methods, page 9). Thus, providing more robust results for memory performance showing an action-emotion interaction following an inverted-U for individuals with putatively normal levels of arousal (Pages 14, 15, 16, 17. Figure 5). We therefore provide more robust results for memory performance showing an action-emotion interaction following an inverted-U for individuals with putatively normal levels of arousal.

“... we performed a final experiment, which we subsequently replicated (Exp 7A and B, respectively). Both were identical to Exp 4 except that, instead of grayscale pictures of objects, participants were presented with an equal number of neutral and emotional color scenes from a standardized database. The cue to Go or NoGo was again indicated by a blue/yellow frame. The enhancing effect of emotion is known to be greater when memory is tested after long (considered to be from 1h to 24h or more) than after short immediate intervals, thus the surprise recognition test was performed after a 24h delay to promote a greater effect of emotion on memory^{51,52}. We first examined memory for participants showing Go-induced memory enhancement for neutral stimuli (21 of the 31 participants in total in Exp 7 A, Figure 5b). This subgroup was selected in view of the results of Exp 6 showing that individuals not showing action-induced memory enhancement may already be at a heightened level of arousal, which could obscure additional memory effects of the emotional nature of stimuli presented at encoding. Strikingly, although this subgroup of participants show Go-induced encoding enhancement for neutral stimuli, this is not observed for emotional stimuli (Figure 5b). The Go-induced decrease in encoding of emotional pictures is in keeping with our predictions that the combination of emotion and Go-responses moved LC activity beyond the optimum of the inverted-U function for memory encoding (Figure 5a). This effect was replicated in Exp7 B on examining memory for 18 of the 33 participants showing Go-induced

memory enhancement for neutral stimuli (Figure 5d). In both Exp 7A and B, an emotion (neutral, aversive) by response (Go, NoGo) ANOVA on encoding performance showed a significant interaction (Exp 7 A: $F_{1,20} = 7.96$; $P = 0.011$, $\eta^2 = 0.285$; Exp 7 B: $F_{1,17} = 14.795$; $P = 0.001$, $\eta^2 = 0.465$) and a significant main effect of response (Exp 7 A: $F_{1,20} = 5.393$; $P = 0.031$; $\eta^2 = 0.212$; Exp 7 B: $F_{1,17} = 5.257$; $P = 0.035$; $\eta^2 = 0.236$), whereas the main effect of emotion was not significant. Post-hoc t -tests revealed significantly different memory performance between Go Emotional and NoGo Neutral stimuli (Exp 7 A: $t_{20} = 2.881$; $P = 0.009$; Exp 7 B: $t_{17} = 3.374$; $P = 0.004$) and NoGo Neutral and NoGo Emotional stimuli (Exp 7 A: $t_{20} = -2.598$; $P = 0.017$; Exp 7 B: $t_{17} = -2.790$; $P = 0.0134$). The difference between Go vs. NoGo Neutral stimulus encoding (Exp 7 A: $t_{20} = 6.622$; $P < 0.001$; Exp 7 B: $t_{17} = 5.924$; $P < 0.001$) is obviously biased by preselection of participants showing this effect.

Interestingly, those participants that do not show Go-induced memory enhancement for neutral stimuli (Exp 7 A: $N = 10$, Exp 7 B: $N = 15$), actually show better memory for NoGo neutral pictures (Figure 5c, e). If a Go-induced release of NE impairs memory in these participants, this would be compatible with these subjects operating more to the right of the inverted-U function of arousal, *i.e.*, they were in a state of higher arousal than other subjects during the course of the experiment (Figure 5c, e), in line with our findings from subjects in Exp 6 showing attenuated light-reflex. In Exp 7A, we again find an opposite pattern for Go/NoGo effects on emotional stimuli (Figure 5c), and a significant interaction between emotion and motor response ($F_{1,9} = 48.171$; $P < 0.001$; $\eta^2 = 0.843$). This interaction was not, however, found for Exp 7 B ($F_{1,14} = 2.714$; $P = 0.122$). Nevertheless, the results of Exp 7 – overall – indeed confirm our predictions, based on the Yerkes-Dodson law, for memory performance showing an action-emotion interaction following an inverted-U for individuals with putatively normal levels of arousal. Moreover, they provide further support for a NE basis of action-induced memory enhancement.”

In the Methods section, we now provide the updated subject numbers for Exp 7A and B (Page 9-10)

“ Exp 7.

Participants performing at less than 85% correct button press for Go, and 85% correct withheld responses for NoGo, trials were excluded from analyses. Furthermore, those participants with poor memory performance for collapsed emotional and neutral stimuli or for emotional or neutral stimuli respectively (defined as correct hit remembered rate minus remember false alarm rate less than 0%) were not further considered for analysis.

Exp 7 A: 38 participants (20 females; 36 right handed; age range, 18–35 years; mean age, 28.85; SD, 6.22) performed this experiment. The same presentation time and ISI as in Exp 4-6 were used for both encoding and recognition phases.

One was excluded from further analysis on the basis of Go/NoGo performance and six participants due to poor memory performance.

Exp 7 B: This experiment was performed as a replication of Exp 7 A. In the former subjects were included in the study until reaching an effect size of interest (interaction between emotion and action for subjects that show AIME) of at least 25%. For Exp 7 B the same stop criteria for including subjects in the study was applied preserving a similar sample size as in Exp 7 A..

51 participants (32 females; 47 right handed; age range, 18–35 years; mean age, 25.56; SD, 3.85) performed this experiment. The same presentation time and ISI as in Exp 4-6 were used for both encoding and recognition phases.

Nine were excluded from further analysis on the basis of Go/NoGo performance and fourteen participants due to poor memory performance.”

Figure 5. Go-induced encoding enhancement is modulated by emotion. (a) Schematic “inverted-U” relationship between encoding performance and norepinephrine (NE) level, with putative locus of Go and NoGo encoding for emotionally neutral stimuli indicated on this curve. We hypothesized that emotion would shift memory scores to the right. (b-c) Recognition memory for remembered items (R) corrected for false alarm rates for Go and NoGo neutral and emotional trials (left) and the schematic (right) for

participants that show Go-induced memory enhancement for the neutral stimuli ($N = 16$) (b) and those that do not ($N = 5$) (c) * ($P < 0.05$), ** ($P < 0.01$), * ($P < 0.001$).**

4. How many voxels were significant in the LC cluster presented in Figure 3 and how many overlapped with the LC atlas template ROI?

The following has been added (in red) to the manuscript Results section (Pages 10-11)

“Testing for an interaction between motor response (Go vs. NoGo) and subsequent memory (R vs. F) identified a significant activation in dorsal pons (2 significant voxels), in an area consistent with LC (Figure 3a, Supplementary Table 5). Note that this effect was also observed if the sample was restricted to the 14 subjects showing go-induced memory enhancement. To increase the robustness of spatial localization of this response to LC, we repeated this analysis using an infra-tentorial template for spatially unbiased, nonlinear normalization of brainstem and cerebellum (SUIT) to provide more accurate intersubject-alignment of the brainstem than whole-brain methods. A significant action by subsequent memory interaction was again observed in dorsal pons. **The overlap of this activation (functional image resolution of 2mm isotropic voxels) with a probabilistic atlas of the LC (image resolution of 1mm isotropic voxels) was nine 1mm voxels (Figure 3b).**”

5. For the pupil dilation experiment, it is hard to map the reported significant interaction of response type and subsequent memory onto Figure 3i, as the mean in each condition is not clear from the figure.

Our apologies for the confusing color scheme. Please find changes in Figure 3i addressing this problem. We also moved the dots to the right to better show the error bars pertaining to the residual error of the model.

6. Were pupil measures collected in any of the other experiments? It would have been helpful to collect them in all experiments.

We only recorded pupil diameter in Exp 6. We agree that these measurements would have been valuable for all Exps but in general we have acquired an extensive multi-modal dataset for this study. We agree with Reviewer 1 that a future study including multiple measures of physiological arousal (pupil diameter, skin conductance) would be an interesting avenue to pursue.

7. The neutral condition from Experiment 7 should be reported on in Table 2, with all 21 participants included. Currently it is not clear if the effect replicated in this experiment without omitting those participants who did not show the effect.

The neutral condition for Exp 7 A and Exp 7 B have been updated in table 2.

Table 2. Summary of paired *t*-test results comparing remember accuracy (% remembered items minus false alarm rate) for Go vs. NoGo stimuli for Exp 1-7.

Exp	t -test Go vs. NoGo	P value	Cohen's d Go vs. NoGo
1	$t_{30} = 2.40$	0.023	0.279
2	$t_{37} = 3.28$	0.002	0.566
3	$t_{25} = 2.85$	0.009	0.373
4	$t_{21} = 2.26$	0.034	0.509
5	$t_{20} = 1.41$	0.175	0.293
6	$t_{27} = 2.75$	0.010	0.397
7 A (neutral)	$t_{30} = 0.85$	0.40	0.172
7 B (neutral)	$t_{32} = 0.43$	0.67	0.097

8. Were any participants in more than one of these experiments?

No, given that it is a memory task we could not include same participant in different experiments because we use the same images set for all of them. In addition, the surprise recognition task would no longer be possible. We now state at the beginning of the methods section:

“Participants. A total of 296 human subjects (aged 18-35; 116 female) were recruited via advertisement to participate in our study, which comprised 8 experiments. No individual performed more than one experiment.” (Online Methods Page 1)

9. There were a high number of excluded participants, especially for the fMRI study.

The incidental nature of the encoding task (i.e., the recognition test was a surprise) and brief presentation time make this a difficult memory task so several subjects performed below chance at recognition. The presentation time was reduced from 1s to 250 ms to enable staggering of object picture presentation with respect to Go/NoGo cue. This shorter presentation time – which also reduced memory performance relative to 1s presentation – was maintained for the functional MRI study to eschew saccade-related confounds.

Data from all excluded subjects (and the reason for their exclusion) is provided in the database to be made publicly available. Note that for the Exps in which we report a significant effect on comparing recollection for Go and NoGo items (Exp 1-4 and 6), this difference remains significant (at an alpha of 0.05) if these excluded subjects are included in the paired samples t-test. This is also the case when testing the interaction between emotion and action in Exp 7 A and B (selecting subjects that showed better memory for neutral stimuli paired with Go vs. NoGo trials). This interaction remains significant including the excluded subjects (for Exp 7A, $F_{1,24} = 9.854$, $P = 0.004$, $\eta^2 = 0.291$ and for Exp 7B, $F_{1,27} = 24.444$, $P < 0.001$, $\eta^2 = 0.475$). We do not make reference to these results in our new submission, as the interested reader can easily calculate them from the behavioral data to be linked to the article.

Regarding to the fMRI experiment in particular, MRI scanning has been show to increase sympathetic nervous system activity¹ and cortisol levels². We have shown that subjects that are generally more aroused (reduced light reflex – Exp 6) show less action-induced memory enhancement. Even if we were to repeat the experiment, it remains possible that the AIME would not be as robust as under less stressful conditions.

10. On p. 31, it is reported that “Up to 52 participants performed the experiment”. Why not report the exact number?

The exact number is actually 52 and this has now been corrected in the methods section. (Online Methods Page 3)

11. On p. 32, it is stated that all participants were paid 20 Euros. Does this mean that the performance-based payments the participants were instructed about were not implemented? If so, this deception might have been reported to friends participating in the study, reducing incentives to aim for those stated rewards. Or was it that everyone made it to the maximal payment value?

This is a fair point. We recruited participants via advertisement at the University but normally when we identified people coming in groups we were careful in instructing them not to disclose anything about the task to the others. More generally, we had to ensure that participants did not reveal to others that there would be a surprise recognition test. Specifically, for Exp 3, we instructed and trained subjects to be as fast as they could and to try to not commit errors but in the end they received the full amount for Spanish Science Ministry bureaucratic reasons (cash payments to study participants cannot be made so we had to use vouchers of a fixed amount of money).

We performed a within-study analysis to address this potential confound, which is now included in the manuscript on (Online Methods Page 4) (the plots provided below have not been included in the revised manuscript:

“That all participants were paid the full 20 Euros raised a possibility that subjects informed one another of the fixed financial compensation. We therefore divided subjects in Exp 3 in two groups, early and late (based on order of performing the Exp) and compared RTs and commission errors between these two groups. The rationale here is that if collusion had indeed occurred, later subjects would have been slower and made more commission errors. There were no significant differences in RTs ($t_{12} = 0.96$; $P = 0.351$) or commission error rates ($t_{12} = -0.433$; $P = 0.673$).”

Figure Rebuttal 5. Mean RT and commission errors in Exp 3, with participants grouped into early and late according to order of task completion.

12. On p. 32, I did not follow the statement, “6 other participants were rejected because they presented less than 85% of correct Go or NoGo trials or answers at recognition phase”

We apologise for the poor wording. This has been corrected to:

“6 other participants were rejected because they either made button presses to less than 85% of Go items during encoding, or withheld responses to less than 85% NoGo trials at encoding, or made less than 85% of button press responses to all stimuli during the recognition test.”(Online Methods Page 5)

13. It would be helpful if the authors would address whether their data and materials are ready to be made available in a publicly accessible online repository such as Open Science Framework upon publication, with data from excluded participants included in the files. Making the data available allows other scientists to reproduce the analyses and making the materials available allows for replications with new samples. This is especially important for potentially high profile papers like the present one.

We thank the reviewer for raising this important point. As stated above, all behavioral data have been included in an excel database.

*The DATA AVAILABILITY STATEMENT was updated in the manuscript to the following:
Methods, including statements of data availability and any associated accession codes and references will be available on a public repository upon publication.*

Have all measures and tasks the participants completed been reported?

Yes. We recruited a total of 403 subjects of which 323 were included for the 8 experiments reported here. The results reported in the manuscript pertain to the ones that were included and the rest (the excluded ones) are also reported in the database (ParticipantsDatabase.xlsx) provided in this response to the reviewer's comments.

References

- 1 Muehlhan, M. *et al.* Enhanced sympathetic arousal in response to fMRI scanning correlates with task induced activations and deactivations. *PLoS one* **8**, e72576 (2013).
- 2 Tessner, K. D., Walker, E. F., Hochman, K. & Hamann, S. Cortisol responses of healthy volunteers undergoing magnetic resonance imaging. *Human brain mapping* **27**, 889-895 (2006).
- 3 Murphy, P. R., O'Connell, R. G., O'Sullivan, M., Robertson, I. H. & Balsters, J. H. Pupil diameter covaries with BOLD activity in human locus coeruleus. *Human brain mapping* **35**, 4140-4154, doi:10.1002/hbm.22466 (2014).
- 4 Joshi, S., Li, Y., Kalwani, R. M. & Gold, J. I. Relationships between Pupil Diameter and Neuronal Activity in the Locus Coeruleus, Colliculi, and Cingulate Cortex. *Neuron* **89**, 221-234, doi:10.1016/j.neuron.2015.11.028 (2016).
- 5 Bradley, M. M., Miccoli, L., Escrig, M. A. & Lang, P. J. The pupil as a measure of emotional arousal and autonomic activation. *Psychophysiology* **45**, 602-607 (2008).
- 6 De Berker, A. O. *et al.* Computations of uncertainty mediate acute stress responses in humans. *Nature communications* **7** (2016).
- 7 Leclercq, V. & Seitz, A. R. Fast task-irrelevant perceptual learning is disrupted by sudden onset of central task elements. *Vision research* **61**, 70-76 (2012).
- 8 Leclercq, V., Le Dantec, C. C. & Seitz, A. R. Encoding of episodic information through fast task-irrelevant perceptual learning. *Vision research* **99**, 5-11 (2014).
- 9 Varazzani, C., San-Galli, A., Gilardeau, S. & Bouret, S. Noradrenaline and dopamine neurons in the reward/effort trade-off: a direct electrophysiological comparison in behaving monkeys. *Journal of Neuroscience* **35**, 7866-7877, doi:10.1523/JNEUROSCI.0454-15.2015 (2015).
- 10 Richer, F. & Beatty, J. Contrasting effects of response uncertainty on the task-evoked pupillary response and reaction time. *Psychophysiology* **24**, 258-262 (1987).
- 11 Diamond, D. M., Campbell, A. M., Park, C. R., Halonen, J. & Zoladz, P. R. The temporal dynamics model of emotional memory processing: a synthesis on the neurobiological basis of stress-induced amnesia, flashbulb and traumatic memories, and the Yerkes-Dodson law. *Neural plasticity* **2007** (2007).
- 12 Yerkes, R. M. & Dodson, J. D. The relation of strength of stimulus to rapidity of habit-formation. *Journal of comparative neurology and psychology* **18**, 459-482 (1908).
- 13 Broadhurst, P. L. Emotionality and the Yerkes-Dodson law. *Journal of experimental psychology* **54**, 345 (1957).

Email correspondence with Reviewer 2

On 1/18/18, 1:20 PM, "Bryan Strange" <bryan.strange@upm.es> wrote:

Dear xxx,

Thank you very much for your helpful and highly constructive peer review of our manuscript "Action boosts episodic memory encoding in humans via engagement of a noradrenergic system.", under consideration at Nature Communications. I gather from the editor handling our manuscript, Mary Elizabeth Sutherland, that you are willing to answer a specific question regarding your review. This is again very helpful and is a great example of how signed peer review benefits authors. We will, of course, make reference to our discussion in the rebuttal, so that the editor is aware of our correspondence. I have copied your comments on our manuscript below for your convenience.

Our specific question pertains to the new experiment you suggest we perform in order to disentangle target detection from action. As you correctly state, in our 7 experiments, we deliberately equated probability of occurrence of Go and NoGo items to differentiate our task from those examining target detection and memory (and also to eschew a potential von Restorff effect for infrequent Go items). The dissociation of target detection from action is difficult, given that the behavioral read-out indicating that a target has actually been detected will require the experimental subject to make some sort of response. From our understanding of the experiment you suggest, the current frame color-to-action mapping is replaced with another stimulus-action mapping such as "L" to left-hand button press and "R" for right-hand press. However, it could be argued that these "L" and "R" stimuli would still be "targets" in the same sense as a colored-frame is a target. There would simply now be two types of targets. Before moving forward with another experiment, we thought we would relay this concern to you, particularly if we have misunderstood something. In any case, with the design you mention, we would have to include 2 "NoGo" cue letters, otherwise the "no button-press" condition becomes relatively infrequent, which could bias memory encoding.

Another possible approach to dissociate the memory effects of action from target detection could be a think-Go and think-NoGo experiment (i.e. without any requirement for action) with simultaneous EEG recordings. This task has been shown to elicit N2 and P3 without any overt movement. This way, instead of having a behavioral read-out of target detection we have an ERP read-out. This of course has its own problems in interpretation. Thinking of making an action may engage LC (and is impossible to test in non-human primates) and, conversely, there may be some covert movement on thinking about Go-ing. This approach would require an entire series of experiments, with different control conditions, and would probably be more suitable as a follow-up publication.

Returning to your suggestion, while a new experiment with 2 different cues to move may address the issue of action vs. target detection to some degree, we thought that a new analysis of our existing data might additionally rule out the contribution of low probability-associated memory effects potentially driving the target detection effects reported by Seitz. In this new analysis, we examine memory for Go responses in the context of their local probability of occurrence. Although the global probability of Go and NoGo stimuli is 0.5:0.5, our randomized presentation order leads to variable number of NoGo stimuli preceding each Go. We have extracted the percent subsequently remembered Go items depending on whether there are 0,1,2 or 3 preceding NoGo items. A target detection mechanism underlying action-induced memory enhancement would predict that memory for Go items would increase with increasing preceding number of NoGo items, as put forward by Seitz and others using infrequent targets. We did not observe this relationship. A one-way ANOVA on Go memory with number of consecutive preceding NoGo trials (0 to 3) before a Go trial as a factor did not show a significant effect in any experiment (1-6) (all $P_s > 0.19$). This lack of Go-item memory modulation on the basis of preceding NoGo trials is evidence against a target detection mechanism underlying better memory observed for stimuli paired with Go trials.

We would therefore be very grateful for your thoughts on our issues with the design of the additional experiment you suggest, or whether you think the additional analysis we have now done resolves the matter.

With best wishes,
Bryan

On 23-01-2018 03:18, xxx wrote:

Dear Bryan,

I've been at a conference and so just got a chance to read your email. I agree it is a challenge to disentangle target detection from action. It is taking me some thought to wrap my head back into your design, but I think I was not clear enough in my suggestion. I was trying to figure out how to dissociate action from target processing and thought it could be possible to make the action unrelated to the content of the stimuli. In other words, subjects could be instructed on what action to take before each trial. This would require a brief instruction screen before each image (e.g., "L", "none", "R"), with subjects instructed to take the requested action (or no action) when they see the subsequent image. If action itself is driving the enhanced memory, then pictures seen when pressing "R" or "L" buttons should be remembered better than those seen when no button press is required. The target-stimulus mapping would not happen at the same time as the picture processing (i.e., the cue "L" would be a target indicating use of the left hand and it would be processed before the picture were shown).

The analysis you report is certainly consistent with the action account but as a null effect not so strong... it is a challenge here because your account predicts a null effect of manipulations of target percentage. On the other hand, (just brainstorming here) would your account predict even better memory for pictures shown when more (rather than less) action is taken? I'm not sure I have the best ideas on how to disentangle the action vs. target accounts but think that your findings would have a stronger impact if you can demonstrate that it is taking action that is driving the effects.

Best,
xxx

On 1/26/18, 2:19 PM, "Bryan Strange" <bryan.strange@upm.es> wrote:

Dear xxx,
Thank you once again for your constructive input. We had also brainstormed in the direction of taking more, rather than less, action to potentially increase the memory enhancement. Indeed, this would be predicted by Sebastian Bouret's non-human primate recordings showing increased LC firing with effort of action. We will run a new experiment with 3 conditions - NoGo, Go as fast as you can, Go as fast and as hard as you can - and are currently putting the apparatus together to measure force of button-press. Looking forward to your next round of comments.
Best,
Bryan

On 30-01-2018 05:58, xxx wrote:

Sounds promising - best wishes for the study!

Reviewers' comments:

Reviewer #4 (Remarks to the Author):

Since my role as third reviewer primarily concerns the question as to whether the authors have addressed the issues raised by the previous round of reviews, I will discuss these points first before adding some issues of my own.

(1) The indirect nature of the measurements is somewhat a necessity for LC, given the breadth of experiments, I think this is fine if mentioned. The added sentence seems sufficient to address this.

(2) While I do not doubt the reliability of the results per se, I think it is strange to define a stop criterion based on effect size. Is there any statistical justification for this? Imagine an effect that fluctuates a lot between individuals, wouldn't a stop criterion based on effect size overestimate the true effect size in any case? The argument has been made for p-values, and I think it applies to effect size measures, even if they do not depend on the subject number (provided the same true effect), see, e.g., Wagenmakers, E.J. (2007). *Psychonomic Bulletin & Review*, 14(5), 779-804.

(3) Re. exp. 6 see below.

(4) Although measuring pupil size and fMRI simultaneously is certainly possible and the combination of pupillometry/eye tracking nowadays available at many sites, I agree with the authors that this is not easy to set up, if not yet available at the scanning facility used.

(5) I think experiment 8 makes an interesting addition, but I would leave it to the editor (and maybe the judgement of reviewer 2) as to whether they should be included in the paper that already consists of many distinct experiments.

Two additional comments

(6) As there is an equal number of targets and foils (Rhits-RFA) is an acceptable measure of performance, though d' would seem more natural. It should be mentioned somewhere that this measure (only) makes sense because of this equality. It is also a bit strange (though it should not matter much to the overall results) that subjects are effectively "rejected" (excluded) based on criterion (Unless I misunderstand something, someone with 86% hits and 86% correct rejections would be included, while someone with 100% hits and 84% correct rejections would be excluded, although the performance of the latter subject would be better).

(7) In the context of experiment 6, I am surprised that no reference is made to studies that relate episodic memory to pupil dilation, in part with very similar paradigms (sometimes providing alternative explanations), for example:

Kafkas A, Montaldi D (2011) Recognition memory strength is predicted by pupillary responses at encoding while fixation patterns distinguish recollection from familiarity. *Q J Exp Psychol (Hove)* 64(10):1971–1989.

Naber M, Frässle S, Rutishauser U, Einhäuser W (2013) Pupil size signals novelty and predicts later retrieval success for declarative memories of natural scenes. *J Vis* 13(2):11. doi:10.1167/13.2.11

The two studies cited in this contexts (refs. 43 and 44) use quite different paradigms. The interpretation as arousal is also difficult, as this would probably affect the baseline, which seems to be normalized out in this analysis (the results of exp.7 notwithstanding).

Reviewer #5 (Remarks to the Author):

In this multi-experiment manuscript, the authors performed a close investigation of the potential role of the LC-NE system in modulating episodic memories associated with (rather benign) actions. Specifically, people viewed objects that were surrounded by colored frames, with different colors signifying either "go" or no-go." The action to perform in "go" trials was a simple button-press, which has little memorial content of its own. Despite this rather minimal action requirement, objects viewed during "go" trials showed superior recognition memory in nearly all experiments. These action-requiring trials were found to elicit greater LC-NE activity both in fMRI (experiment 5) and pupillometry (experiment 6). Based on Experiment 6, the authors suggested that tonic arousal during encoding could modulate the action-based contributions of the LC system to memory. This was supported by examining pupil dilation during encoding, suggesting that participants showing higher tonic arousal were less likely to experience an LC boost for action. Experiments 7A and 7B expanded upon this hypothesis, testing memory for emotionally valenced images.

Taken together, this is quite a compelling set of experiments. Although I was not among the original reviewers, I carefully read the authors' summary and responses to the prior reviews, finding their revisions both thoughtful and complete. Naturally, when theorizing about LC-NE activity, there will always be a certain degree of indirect evidence, given the imaging challenges. In my opinion, these experiments converge on an LC contribution to memory performance about as convincingly as one could reasonably hope. I found the behavioral contribution surprising, seeing that such a minimal (and redundant) action could increase visual memory. The authors' theorizing about optimal arousal levels is provocative and interesting. Bottom line: This is a strong and interesting manuscript that should be published. I anticipate it will generate considerable interest.

I have only a couple minor comments:

1) There have been a few recent articles by Unsworth and Robison that address the theoretical relationship between tonic LC-NE arousal and cognitive control, including memory formation. These citations would give the authors a bit more support for their theoretical arguments in the discussion section. Here are two representative articles:

Unsworth, N., & Robison, M.K. (2017). A locus coeruleus-norepinephrine account of individual differences in working memory capacity and attention control. *Psychonomic Bulletin & Review*, 24, 1283–1311.

Unsworth, N., & Robison, M.K. (2017). The importance of arousal for variation in working memory capacity and attention control: A latent variable pupillometry study. *Journal of Experimental Psychology: Learning, Memory, and Cognition*, 43, 1962–1987.

Adding some discussion of this work would bolster the current theoretical arguments.

2) One minor consideration is that the authors occasionally reason from null results. For example, if idea X were correct, there should be an interaction of AxB, and no such interaction was observed. Although I do not consider it necessary to run Bayesian analyses to estimate the "strength" of these null results, some cautionary note should be included.

Overall, this is an excellent project and a solid manuscript.

Reviewers' comments:

Reviewer #4 (Remarks to the Author):

Since my role as third reviewer primarily concerns the question as to whether the authors have addressed the issues raised by the previous round of reviews, I will discuss these points first before adding some issues of my own.

The indirect nature of the measurements is somewhat a necessity for LC, given the breadth of experiments, I think this is fine if mentioned. The added sentence seems sufficient to address this.

We thank the reviewer for appreciating the necessities of LC research in the context of human studies.

(2) While I do not doubt the reliability of the results per se, I think it is strange to define a stop criterion based on effect size. Is there any statistical justification for this? Imagine an effect that fluctuates a lot between individuals, wouldn't a stop criterion based on effect size overestimate the true effect size in any case? The argument has been made for p-values, and I think it applies to effect size measures, even if they do not depend on the subject number (provided the same true effect), see, e.g., Wagenmakers, E.J. (2007). *Psychonomic Bulletin & Review*, 14(5), 779-804.

The reviewer makes an entirely valid point, which we now address with an additional bootstrap procedure. The following has been added to the manuscript:

"Using a stop criterion based on effect size has its limitations⁴. For this reason, we further validated the statistical robustness of our results by applying a boot-strap procedure of 1000 iterations to the memory data in Exp 7A and 7B, using the MATLAB Resampling statistical toolkit " (Page 44).

Please find new green font insets in pages 17, 18 and 19:

"This effect was replicated in Exp7 B on examining memory for 18 of the 33 participants showing Go-induced memory enhancement for neutral stimuli (Figure 5d). In both Exp 7A and B, an emotion (neutral, aversive) by response (Go, NoGo) ANOVA on encoding performance showed a significant interaction (Exp 7 A: $F_{1,20} = 7.96$; $P = 0.011$, $\eta^2 = 0.285$, $P_{Bootstrap1000} = 0.0075$; Exp 7 B: $F_{1,17} = 14.795$; $P = 0.001$, $\eta^2 = 0.465$, $P_{Bootstrap1000} = 0.001$) and a significant

main effect of response (Exp 7 A: $F_{1,20} = 5.393$; $P = 0.031$, $P_{Bootstrap1000} = 0.04$; $\eta^2 = 0.212$; Exp 7 B: $F_{1,17} = 5.257$; $P = 0.035$; $\eta^2 = 0.236$, $P_{Bootstrap1000} = 0.053$), whereas the main effect of emotion was not significant. Note that a bootstrap procedure was applied to the statistical model estimation for Exp 7 A and B, given that the sample size of the replication study was based on effect size (see Online Methods). Post-hoc t -tests revealed significantly different memory performance between Go Emotional and NoGo Neutral stimuli (Exp 7 A: $t_{20} = 2.881$; $P = 0.009$; Exp 7 B: $t_{17} = 3.374$; $P = 0.004$) and NoGo Neutral and NoGo Emotional stimuli (Exp 7 A: $t_{20} = -2.598$; $P = 0.017$; Exp 7 B: $t_{17} = -2.790$; $P = 0.0134$). The difference between Go vs. NoGo Neutral stimulus encoding (Exp 7 A: $t_{20} = 6.622$; $P < 0.001$; Exp 7 B: $t_{17} = 5.924$; $P < 0.001$) is obviously biased by preselection of participants showing this effect.

Interestingly, those participants that do not show Go-induced memory enhancement for neutral stimuli (Exp 7 A: $N = 10$, Exp 7 B: $N = 15$), actually show better memory for NoGo neutral pictures (Figure 5c, e). If a Go-induced release of NE impairs memory in these participants, this would be compatible with these subjects operating more to the right of the inverted-U function of arousal. This would imply they were in a state of higher arousal than other subjects during the course of the experiment (Figure 5c, e), in line with our findings from subjects in Exp 6 showing attenuated light-reflex. In Exp 7A, we again find an opposite pattern for Go/NoGo effects on emotional stimuli (Figure 5c), and a significant interaction between emotion and motor response ($F_{1,9} = 48.171$; $P < 0.001$, $P_{Bootstrap1000} = 0.0005$; $\eta^2 = 0.843$). This interaction was not, however, found for Exp 7 B ($F_{1,14} = 2.714$; $P = 0.122$, $P_{Bootstrap1000} = 0.121$). Nevertheless, the results of Exp 7 – overall – indeed confirm our predictions, based on the Yerkes-Dodson law, for memory performance showing an action-emotion interaction following an inverted-U for individuals with putatively normal levels of arousal. Moreover, they provide further support for a NE basis of action-induced memory enhancement.”

(3) Re. exp. 6 see below.

(4) Although measuring pupil size and fMRI simultaneously is certainly possible and the combination of pupillometry/eye tracking nowadays available at many sites, I agree with the authors that this is not easy to set up, if not yet available at the scanning facility used.

We thank the reviewer for her/his understanding.

(5) I think experiment 8 makes an interesting addition, but I would leave it to the editor (and maybe the judgement of reviewer 2) as to whether they should be included in the paper that already consists of many distinct experiments.

We are pleased that the reviewer considers our Exp 8 interesting. While we leave the issue of inclusion of data from Exp8A, B and C to the editor, we remain of the opinion that these results are sufficiently extensive so as to warrant writing up in a further manuscript focused on action selection and memory. There are additional points to be made regarding these results, **[Redacted]**. Thus, a comprehensive account of the memory effects present in Exp 8 requires, in our opinion, an additional paper. Including these results in the current manuscript would make thinks really rather long.

Two additional comments

(6) As there is an equal number of targets and foils (Rhits-RFA) is an acceptable measure of performance, though d' would seem more natural. It should be mentioned somewhere that this measure (only) makes sense because of this equality.

The reviewer is correct here, and our equal target:foil ratio was indeed our motivation for using the Rhits-RFA measure.

The following has been added to the methods section:

“Our equal target:foil ratio allows as to define memory performance as correct hit remembered rate minus remember false alarm rate. Those participants with memory performance less than 0% were not further considered for analysis.” (Page 35)

It is also a bit strange (though it should not matter much to the overall results) that subjects are effectively “rejected” (excluded) based on criterion (Unless I misunderstand something, someone with 86% hits and 86% correct rejections would be included, while someone with 100% hits and 84% correct rejections would be excluded, although the performance of the latter subject would be better).

We apologize for any misunderstanding here. Our exclusion criterion is % of hits – fa > 0; i.e., someone with more false alarms than hits would be rejected as they are performing below chance.

someone with 86% hits and 86% correct rejections would be included

True. Since fa = 1-cr that would imply fa = 14% if hits are 86% then hits-fa = 72% → Included

someone with 100% hits and 84% correct rejections would be excluded

Not necessarily. Fa = 100-84 = 16% → criterion = 100-16 = 84% → Included

(7) In the context of experiment 6, I am surprised that no reference is made to studies that relate episodic memory to pupil dilation, in part with very similar paradigms (sometimes providing alternative explanations), for example:

Kafkas A, Montaldi D (2011) Recognition memory strength is predicted by pupillary responses at encoding while fixation patterns distinguish recollection from familiarity. *Q J Exp Psychol* (Hove) 64(10):1971–1989.

Naber M, Frässle S, Rutishauser U, Einhäuser W (2013) Pupil size signals novelty and predicts later retrieval success for declarative memories of natural scenes. *J Vis* 13(2):11. doi:10.1167/13.2.11

The two studies cited in this contexts (refs. 43 and 44) use quite different paradigms.

We thank the reviewer for pointing us towards these papers, which we now cite on page 13:

“Furthermore, pupil diameter is also modulated by learning and memory processes^{43,44}.”

The interpretation as arousal is also difficult, as this would probably affect the baseline, which seems to be normalized out in this analysis (the results of exp.7 notwithstanding).

The reviewer is correct in noting that arousal levels also influence baseline pupil diameter. In the case of our behavioural paradigm, however, stimulus onset asynchrony is too short for us to be sure that the pupil returns to baseline diameter before the next stimulus appears. For this reason, we elected to measure changes in light reflex as an index of arousal. The following has

been added to the results section (Action-induced episodic memory enhancement depends on arousal)

“To confirm this prediction, we calculated the light-reflex amplitude for all participants in Exp 6 by averaging across all encoding trials for each subject. We note that measuring baseline pupil diameter would also provide an index of arousal, but we elected to measure changes in light reflex instead because the interstimulus interval in the current task may not have been sufficient for the pupil to return to baseline diameter prior to each stimulus.” (Page 15).

Reviewer #5 (Remarks to the Author):

In this multi-experiment manuscript, the authors performed a close investigation of the potential role of the LC-NE system in modulating episodic memories associated with (rather benign) actions. Specifically, people viewed objects that were surrounded by colored frames, with different colors signifying either "go" or no-go." The action to perform in "go" trials was a simple button-press, which has little memorial content of its own. Despite this rather minimal action requirement, objects viewed during "go" trials showed superior recognition memory in nearly all experiments. These action-requiring trials were found to elicit greater LC-NE activity both in fMRI (experiment 5) and pupillometry (experiment 6). Based on Experiment 6, the authors suggested that tonic arousal during encoding could modulate the action-based contributions of the LC system to memory. This was supported by examining pupil dilation during encoding, suggesting that participants showing higher tonic arousal were less likely to experience an LC boost for action. Experiments 7A and 7B expanded upon this hypothesis, testing memory for emotionally valenced images.

Taken together, this is quite a compelling set of experiments. Although I was not among the original reviewers, I carefully read the authors' summary and responses to the prior reviews, finding their revisions both thoughtful and complete. Naturally, when theorizing about LC-NE activity, there will always be a certain degree of indirect evidence, given the imaging challenges. In my opinion, these experiments converge on an LC contribution to memory performance about as convincingly as one could reasonably hope. I found the behavioral contribution surprising, seeing that such a minimal (and redundant) action could increase visual memory. The authors' theorizing about optimal arousal levels is provocative and interesting. Bottom line: This is a strong and interesting manuscript that should be published. I anticipate it will generate considerable interest.

We are very grateful to the reviewer for his/her constructive and encouraging comments.

I have only a couple minor comments:

- 1) There have been a few recent articles by Unsworth and Robison that address the theoretical relationship between tonic LC-NE arousal and cognitive control, including memory formation.

These citations would give the authors a bit more support for their theoretical arguments in the discussion section. Here are two representative articles:

Unsworth, N., & Robison, M.K. (2017). A locus coeruleus-norepinephrine account of individual differences in working memory capacity and attention control. *Psychonomic Bulletin & Review*, 24, 1283–1311.

Unsworth, N., & Robison, M.K. (2017). The importance of arousal for variation in working memory capacity and attention control: A latent variable pupillometry study. *Journal of Experimental Psychology: Learning, Memory, and Cognition*, 43, 1962–1987.

Adding some discussion of this work would bolster the current theoretical arguments.

We now include mention to this work (Pages 16,17)

“Indeed, it is not the first time that different levels of LC-NE tonic functioning have been suggested to regulate other aspects of cognition including working memory. Particularly it has been recently hypothesized that there are different potential LC-NE modes explaining low working memory capacity performance: lower tonic LC activity (those that would be operating in the left side of the curve; Figure 5a), hyperactive tonic LC activity (operating in the right side of the curve) or increased variability in LC tonic activity⁵³.”

2) One minor consideration is that the authors occasionally reason from null results. For example, if idea X were correct, there should be an interaction of AxB, and no such interaction was observed. Although I do not consider it necessary to run Bayesian analyses to estimate the "strength" of these null results, some cautionary note should be included.

The reviewer is correct in pointing to a short-coming in our interpretation of null results.

a. We have now toned this down e.g. saying “We note that this reasoning is derived from a null result” Page 7

b. Although the reviewer did not consider it necessary to run Bayesian analyses, we have now discussed our null results calculating posterior probabilities of the null hypothesis using Bayesian hypothesis testing by means of the BIC approximation approach suggested in¹.

Pages 6, 7, 8, 9, 10 and 35. Supplementary tables 3, 4

“We note that this reasoning is derived from a null result. Since using traditional p value hypothesis testing one can fail to reject the null hypothesis but the null hypothesis can never be accepted, we calculated posterior probabilities of the null hypothesis using Bayesian hypothesis testing²⁴. Bayesian information criterion (BIC)-based estimation of posterior probabilities revealed $\Delta BIC_{10} = 92.04$; $\Delta BIC_{10} = 92.04$; $Pr_{BIC}(H_0|D) \approx 1$ indicating very strong evidence in favor of the null hypothesis (Supplementary Table 3).” Page 7

“The BIC-based estimation of posterior probabilities revealed a $\Delta_{BIC_{10}} = 198.61$; $Pr_{BIC}(H_0|D) \approx 1$, indicating very strong evidence in favor of the null hypothesis.” Page 8

“BIC-based estimation of posterior probabilities yielded a $\Delta_{BIC_{10}} = 92.04$; $Pr_{BIC}(H_0|D) \approx 1$, showing very strong evidence in favor of the null hypothesis.” Page 9

“The BIC-based estimation of posterior probabilities revealed a $\Delta_{BIC_{10}} = 243.42$; $Pr_{BIC}(H_0|D) \approx 1$, which shows very strong evidence in favor of the null hypothesis.” Page 10

“Here, and in subsequent experiments, we make reference to absence of statistically significant effects. Using traditional p value hypothesis testing one can fail to reject the null hypothesis but the null hypothesis can never be accepted. Thus, for each reported null result we additionally provide the posterior probabilities using Bayesian hypothesis testing by means of the BIC approximation approach. When comparing a null hypothesis H_0 with an alternative hypothesis H_1 the difference in BIC values can be written as:

$$\Delta BIC_{10} = n \log \left(\frac{SSE_1}{SSE_0} \right) + \log(n);$$

where SSE_1 is the sums of squared errors for model H_1 . Posterior probability for the null hypothesis H_0 can be written as:

$$Pr_{BIC}(H_0|D) = \frac{1}{1 + \exp(-\frac{1}{2}\Delta BIC_{10})};$$

where a posterior probability between .95 and .99 represents strong evidence and above .99 very strong evidence⁴.” Page 36

Supplementary Table 3

Exp	Commission errors by number of preceding Go stimuli	Linear Contrast	Memory performance for the NoGo stimuli by number of preceding Go stimuli
1	$F_{1.88,56.44} = 1.90; P = 0.162$	$F_{1,30} = 5.44; P = 0.027; \eta^2 = 0.349$	$F_{2,34,70.16} = 0.82; P = 0.487$ $\Delta BIC_{10} = 92.04; Pr_{BIC \sim 1}$
2	$F_{2,1,77.684} = 2.50; P = 0.086$	$F_{1,37} = 4,64; P = 0.038; \eta^2 = 0.111$	$F_{1,93,71.48} = 0.34; P = 0.707$ $\Delta BIC_{10} = 159.55; Pr_{BIC \sim 1}$
3	$F_{1,685,42.13} = 0.87; P = 0.410$	$F_{1,25} = 1.56; P = 0.220, \eta^2 = 0.059$	$F_{2,09,52.17} = 0.36; P = 0.710$ $\Delta BIC_{10} = 113.52; Pr_{BIC \sim 1}$
4	$F_{1,7,35.75} = 2.96; P = 0.072$	$F_{1,21} = 7.79; P = 0.011, \eta^2 = 0.271$	$F_{2,09,43.97} = 1.40; P = 0.260$ $\Delta BIC_{10} = 59.30; Pr_{BIC \sim 1}$
5	$F_{1,741,34,825} = 2,34; P = 0.118$	$F_{1,20} = 3,93; P = 0.061, \eta^2 = 0.164$	$F_{2,53,20.66} = 0.06; P = 0.968$ $\Delta BIC_{10} = 103.75; Pr_{BIC \sim 1}$
6	$F_{1,3,34.97} = 3.55; P = 0.058$	$F_{1,27} = 4.72; P = 0.390, \eta^2 = 0.149$	$F_{1,75,47.38} = 1.42; P = 0.250$ $\Delta BIC_{10} = 143.35; Pr_{BIC \sim 1}$

Supplementary Table 4

Exp	One Way ANOVA on memory performance	BIC-based estimation of posterior probabilities
1	$F_{1,98,59.53} = 1.72; P = 0.188$	$\Delta BIC_{10} = 92.04; Pr_{BIC \sim 1}$
2	$F_{1,40,52} = 0.61; P = 0.49$	$\Delta BIC_{10} = 109.94; Pr_{BIC \sim 1}$
3	$F_{2,2,55} = 0.36; P = 0.72$	$\Delta BIC_{10} = 164.80; Pr_{BIC \sim 1}$
4	$F_{2,05,42.98} = 1.63; P = 0.21$	$\Delta BIC_{10} = 96.39; Pr_{BIC \sim 1}$
5	$F_{2,05,41.07} = 0.16; P = 0.85$	$\Delta BIC_{10} = 103.75; Pr_{BIC \sim 1}$
6	$F_{1,85,49.94} = 0.18; P = 0.82$	$\Delta BIC_{10} = 143.35; Pr_{BIC \sim 1}$

Overall, this is an excellent project and a solid manuscript.

REFERENCES

- 1 Wagenmakers, E.-J. A practical solution to the pervasive problems of values. *Psychonomic bulletin & review* **14**, 779-804 (2007).

REFEREE COMMENTS

Referee #4: Please let me apologize for the delay I caused in the reviewing process. I had mixed this up with another manuscript and thought I had submitted my feedback already. The revision addressed my concerns to my satisfaction, and I have no strong opinion regarding whether exp. 8 should be included, so I would leave this to the authors.

REFeree COMMENTS

Referee #4: Please let me apologize for the delay I caused in the reviewing process. I had mixed this up with another manuscript and thought I had submitted my feedback already. The revision addressed my concerns to my satisfaction, and I have no strong opinion regarding whether exp. 8 should be included, so I would leave this to the authors.

We thank the reviewer for these final comments